# Think Smarter not Harder:
# Adaptive Reasoning with Inference Aware Optimization

**Zishun Yu** [1]  **Tengyu Xu** [2]  **Di Jin** [2]  **Karthik Abinav Sankararaman** [2]  **Yun He** [2]  **Wenxuan Zhou** [2]
**Zhouhao Zeng** [2]  **Eryk Helenowski** [2]  **Chen Zhu** [2]  **Sinong Wang** [2]  **Hao Ma** [2]  **Han Fang** [2]

## Abstract

Solving mathematics problems has been an intriguing capability of language models, and many efforts have been made to improve reasoning by extending reasoning length, such as through self-correction and extensive long chain-of-thoughts. While promising in problem-solving, advanced long reasoning chain models exhibit an undesired uni-modal behavior, where trivial questions require unnecessarily tedious long chains of thought. In this work, we propose a way to allow models to be aware of inference budgets by formulating it as utility maximization with respect to an inference budget constraint, hence naming our algorithm Inference Budget-Constrained Policy Optimization (IBPO). In a nutshell, models fine-tuned through IBPO learn to "understand" the difficulty of queries and allocate inference budgets to harder ones. With different inference budgets, our best models are able to have a $4.14\%$ and $5.74\%$ absolute improvement ($8.08\%$ and $11.2\%$ relative) on MATH500 using $2.16x$ and $4.32x$ inference budgets respectively, relative to LLaMA3.1 8B Instruct. These improvements are approximately 2x those of self-consistency under the same budgets.

## 1. Introduction

Complex reasoning has been an intriguing ability of large language models (LLMs), with application in for example mathematical problem-solving (Cobbe et al., 2021; Hendrycks et al., 2021b; Lightman et al., 2023) or coding (Chen et al., 2021; Austin et al., 2021; Hendrycks et al., 2021a), which does not only require nature language comprehending but also logical and critical "thinking". An observation merged in the LLM reasoning literature is that longer reasoning traces often leads to improved reasoning soundness and correctness. The seminal work of chain-of-thought (CoT) (Wei et al., 2022) is an excellent example of how enriching reasoning details, by decomposing reasoning traces into steps, improves its problem-solving capability. CoT has been considered a standard technique in reasoning, recent works extend CoT by allow LLMs to expand its reasoning steps, by for example CoT with more steps (Jin et al., 2024) (as explicitly required by instruction), self-reflection/correction (Madaan et al., 2024; Zelikman et al., 2022; Yan et al., 2024; Qu et al., 2024), multi-turn reasoning (Kumar et al., 2024) or multi-agent debate (Liang et al., 2023; Pham et al., 2023) (as a heterogeneous case of multi-turn). It was conjectured that scaling the test-time compute or the reasoning length unleashes LLMs' potential for reasoning (Snell et al., 2024), which has been empirically verified by recent hype of ultra-long reasoning models, such as OpenAI-o1 (Jaech et al., 2024) and DeepSeek-R1 (DeepSeek-AI et al., 2025). We'll later categorizes these type of responses as (standard) CoT responses, extended responses, and (ultra-)long responses, respectively, given the nature of their reasoning lengths.

While scaling reasoning length is promising in problem-solving, advanced long reasoning-chain models show an undesired uni-modal behavior that trivial questions may require unnecessarily tedious long reasoning trace, an example is shown in Appendix A. This uni-modal behavior creates unnecessarily higher inference costs and increased carbon footprints (Henderson et al., 2020; Anthony et al., 2020). To partially address this, we study how to enable multi-modal behavior for reasoning models in a way the length of reasoning traces are automatically adjusted according to the hardness level of the queries. From the aspect of query-adaptive reasoning length, some heuristic methods (e.g. Aggarwal et al., 2023; Xu et al., 2024a; Wang et al., 2024) have been making effort towards better token efficiency, by which is meant better accuracy with (hopefully) less token overhead. We take a reinforcement learning (RL) perspective, where the accuracy gain over the token overhead is nothing but a non-differentiable objective to be optimized. One could, for instance, take the negative response length or a metric of this

---

[1]Department of Computer Science, The University of Illinois Chicago, USA [2]MetaAI, Menlo Park, USA. Correspondence to: Zishun Yu <zyu32@uic.edu>.

*Proceedings of the 42nd International Conference on Machine Learning*, Vancouver, Canada. PMLR 267, 2025. Copyright 2025 by the author(s).

sort as an intrinsic reward (Chentanez et al., 2004; Pathak et al., 2017). However, balancing the intrinsic and extrinsic (accuracy) rewards might also be challenging (Liu et al., 2021), and might be vulnerable to reward hacking (Pan et al., 2022; Skalse et al., 2022; Karwowski et al., 2023).

Instead explicit modeling the length of responses, we take a more abstract formulation, where we consider labeling each response $y$ with an unique group label $\mathcal{G}_i : i \in [\![G]\!]$ for total number of $G$ groups, so that the union of these disjoint groups exactly form the response space $\cup_i \mathcal{G}_i = \mathcal{Y}$. For example $\mathcal{G}$ could be the group of CoT responses (with standard length) or extended responses. We could then impose an density constraint on each or a set of groups, by caping $\mathbb{E}_{x \sim \mu} \mathbb{E}_{y \sim \pi(x)}[\mathbb{1}_{\{y \in \mathcal{G}_i\}}] \leq q_i$ for some prompt distribution $\mu$ and some response distribution $\pi(x)$, induced from LLMs, conditioned on a prompt $x$. This naturally formulates a constrained RL (Garcıa & Fernández, 2015; Altman, 2021) problem. Also this group definition is motivated by the resource-allocation literature (Chenery & Kretschmer, 1956; Ibaraki & Katoh, 1988; Karlin, 2003), from the optimization and econometric communities, which have been later applied in many machine learning applications (e.g. Zemel et al., 2013; Badanidiyuru et al., 2018). This generalization allows potential broader application of our algorithm as discussed in Section 6. We've now set our goal:

> A constrained RL framework controlling how response groups $\{\mathcal{G}_i\}$ are distributed.

Therefore, one could control how responses of different lengths (which are supposed to belong to different groups) are distributed. Our rationale of algorithm design is given in Section 2, derived from an optimization perspective but ended as a very simple generalization of iterative supervised fine-tuning (SFT) methods such as reward-ranking fine-tuning (RAFT) (Dong et al., 2023) and rejection sampling fine-tuning (RFT) (Ouyang et al., 2022; Touvron et al., 2023), see details in Section 3. Given the motivation of our algorithm, we call the resulted algorithm as Inference Budget-Constrained Policy Optimization (IBPO).

**Paper structure.** Section 2 present the derivation of our algorithm from an optimization view, resulting in a simple weighted iterative SFT update. Section 3 provides further details on practical implementation, including the base algorithm and the reward design. Section 4 introduces some experimental settings. The empirical results of our IBPO are presented in Section 5. And Section 6 concludes our work with limitations, broader impact, and further discussions.

## 2. Algorithm Design

**Problem setup.** To make the notation compact, we take the sequence-level notation (or the bandit notation), commonly used in LLMs (Ziegler et al., 2019; Rafailov et al., 2024), especially in preference modeling, that suppresses the transition probabilities and intermediate rewards, and see a response as a whole. In particular, a policy $\pi : \mathcal{X} \to \Delta(\mathcal{Y})$ takes a prompt $x \in \mathcal{X}$ and draw a response $a_1 \circ a_2 \cdots \circ a_T =: y \in \mathcal{Y}$ from the produced probability simplex $\Delta(\mathcal{Y})$, where $\circ$ denotes concatenation, $a_i \in \mathcal{V}$ corresponds to the $i$-th token drawn from the vocabulary $\mathcal{V}$, and $T$ is the maximum length. A LLM is a parametric policy $\pi_\theta \in \Pi_\theta \subseteq \Pi$, where $\Pi_\theta$ and $\Pi$ are the parametric and non-parametric policy space respectively. Let $\mathcal{J}(\pi; \mu, r)$ or sometimes $\mathcal{J}(\pi)$ be a general objective function, defined by a prompt distribution $\mu \in \Delta(\mathcal{X})$, and a bounded reward function $r : \mathcal{X} \times \mathcal{Y} \to [-R_{\max}, R_{\max}]$. Also, we define $\mu_\Omega$ as an empirical distribution induced from $\Omega$, a set of prompts.

As aforementioned, we define $G$ disjoint groups $\mathcal{G}_i$ such that $\cup_i \mathcal{G}_i = \mathcal{Y}$ and $\mathcal{G}_i \cap \mathcal{G}_j = \emptyset$ for all $i \neq j$. Each response $y \in \mathcal{Y}$ is attached to exactly one group, in the sense that $y \in \mathcal{G}_i$ for some $i$. In the context of LLM reasoning, without loss of generality, we consider two groups for brevity: $\mathcal{G}_\circ$ and $\mathcal{G}_+$, corresponding to regular-length CoT responses and extended responses (with low and high inference costs), respectively. To conclude the formulation of constrained RL with resource allocation constraints, we could in general define the feasible set as a convex polytope, $\Phi_\mathcal{G} := \{\pi : \mathbb{E}_x \mathbb{E}_{y \sim \pi(x)}[\mathbb{1}_{\{y \in \mathcal{G}_i\}}] \leq q_i \text{ for all } i\}$, that caps the total density mass of each group $\mathcal{G}_i$ by $q_i$. In our setting, we only need to cap the total mass of extended responses to optimize the inference efficiency, posing a half-space $\Phi_+ := \{\pi : \mathbb{E}_x \mathbb{E}_{y \sim \pi(x)}[\mathbb{1}_{\{y \in \mathcal{G}_+\}}] \leq q_+\}$ for some $q_+ > 0$.

**Background.** With the bandit setup, our setting draw a lot connections to the online learning literature, especially online/bandit convex optimization (Hazan et al., 2016; Slivkins et al., 2019), although we optimize a fixed function the training data is however collected in an online fashion. In the sense of distributing resources across groups, it is connected to for example knapsack bandits (Badanidiyuru et al., 2018) and statistical parity (Zemel et al., 2013), aka group fairness. Although the definition of groups and optimization programs could be different. This allocation optimization point of view allows us to further extend our method to broader LLM applications. To the end of policy optimization with constraints, common techniques include projection (Zinkevich, 2003; Flaxman et al., 2004; Bubeck et al., 2015; Yang et al., 2020), Lyapunov-based (Chow et al., 2018; Cayci et al., 2022), or Lagrangian methods (Ray et al., 2019).

**Non-parametric space $\Pi$.** In our case, our goal is to solve:

$$\max_{\pi_\theta \in \Pi_\theta} \mathcal{J}(\pi_\theta) \quad \text{s.t. } \pi_\theta \in \Phi_+ \tag{1}$$

where $\Pi_\theta$ is the parameterized policy space. Solving Eq. (1) is however intractable due to the LLM parameterized policy space $\Pi_\theta$. A common practice is alternating gradients

between reward maximization and constraint satisfaction. For example in Lagrangian methods such as TRPO/PPO-Lagrangian (Ray et al., 2019), one could do alternating update $\pi_\theta$ and the Lagrangian multiplier. Another workaround is to first obtain a solution $\pi^\star$ in the non-parametric policy space $\Pi := \Delta(\mathcal{X} \times \mathcal{Y})$, aka tabular representations, and project $\pi^\star$ onto the parameteric one $\Pi_\theta$, as a technique used in many (constrained) RL works (Peters et al., 2010; Montgomery & Levine, 2016; Zhang et al., 2020).

The advantage of working in the non-parametric space $\Pi$ is: solving $\max_{\pi \in \Pi} \mathcal{J}(\pi)$ s.t. $\pi \in \Phi_+$ is easy, on the conditions that (i) $\mathcal{J}(\pi)$ is concave in $\pi$ so that it is a convex program, and (ii) sampling and evaluating the reward function $r(\cdot, \cdot)$ and the cost indicator $\mathbb{1}_{\{\cdot\}}$ are cheap. The condition (i) is sometimes true, for instances: bandit objective (Slivkins et al., 2019); the LP formulation (Manne, 1960; Denardo, 1970; Nachum & Dai, 2020; Nachum et al., 2019) and (relative) entropy regularized RL (Ziebart, 2010; Haarnoja et al., 2017; 2018) are often concave in occupancy measure $\rho$. However, in rare case condition (ii) holds, RL works (Haarnoja et al., 2018; Peng et al., 2019; Zhang et al., 2020) often resort to value function approximation, making it is easy, for discrete action space, to evaluate $Q$-values (or alternatively advantages) for all actions for a specific state. It is then tractable to obtain closed-form solution (optimal w.r.t. the value/advantage approximations) in $\Pi$. Once an optimal policy $\pi^\star$ is found, one could then project it onto $\Pi_\theta$ through (reverse) information projection $\theta = \arg\min_\theta \mathbb{KL}(\pi^\star \| \pi_\theta)$ (often done approximately by taking gradient steps).

**Stochastic optimization.** With a LLM, it is obviously intractable to sample and evaluate the reward function $r(x, y)$ for all $(x, y) \in \mathcal{X} \times \mathcal{Y}$, similarly for the cost indicator $\mathbb{1}_{\{\cdot\}}$. To avoid the training of additional value models for LLMs (Snell et al., 2022; Yu et al., 2023), which can create significant overhead in terms of memory usage, implementation complexity, and training stability, we consider a stochastic optimization. The stochastic counterpart as described in Eq. (2) solves an approximate $\hat{\pi}^\star$ using a manageable number of samples rather than directly solving for the global optimum $\pi^\star$, still, in the non-parametric space $\Pi$:

$$\hat{\pi}^\star(\mathbf{X}, \mathbf{Y}) \in \arg\max_{\pi \in \Pi} \hat{\mathcal{J}}(\pi; \mathbf{X}, \mathbf{Y}) \text{ s.t. } \pi \in \hat{\Phi}_+(\mathbf{X}, \mathbf{Y})$$
$$\hat{\mathcal{J}}(\pi; \mathbf{X}, \mathbf{Y}) := \frac{1}{nm} \sum_i^n \sum_j^m [\pi(y_{ij}|x_i) r(x_i, y_{ij})]$$
$$\hat{\Phi}_+(\mathbf{X}, \mathbf{Y}) := \{\pi : \sum_i \sum_j [\pi(y_{ij}|x_i)(\mathbb{1}_{\{y_{ij} \in \mathcal{G}_+\}} - q_+)] \leq 0\}$$
$$(2)$$

where $\mathbf{X} \in \mathcal{X}^n$ is a vector of $n$ sampled prompts and $\mathbf{Y} \in \mathcal{Y}^{n \times m}$ is a matrix of responses, with $m$ responses for each of the $n$ prompts; we explicitly write $\hat{\mathcal{J}}$ with the conventional expected reward maximization for notational convenience, though alternative objectives are not restricted.

Since the empirical problem (2) is a convex program with relative small sample size, it is now manageable. Comb-

ing the aforementioned projection step, we could write the program as a bi-level stochastic optimization:

$$\pi_\theta = \arg\min_{\pi_\theta \in \Pi_\theta} \mathbb{E}_x \left[ \mathbb{KL}(\hat{\pi}^\star_{\mathbf{X}, \mathbf{Y}_\theta} \| \pi_\theta)[x] \right]$$
$$\text{s.t. } \hat{\pi}^\star_{\mathbf{X}, \mathbf{Y}_\theta} \in \arg\max_{\pi \in \Pi \cap \hat{\Phi}_+(\mathbf{X}, \mathbf{Y}_\theta)} \hat{\mathcal{J}}(\pi; \mathbf{X}, \mathbf{Y}_\theta)$$
$$(3)$$

where $\mathbf{Y}_\theta \sim \pi_\theta(\mathbf{X})$, hence $\hat{\pi}^\star$ is indirectly a function of $\theta$.

**Practical update.** For general bi-level optimization problems, iteratively solving the upper and lower-level problems by alternatively fixing one while optimizing the other could be expensive (Zhang et al., 2024). We've however already setup a manageable inner problem, making it easy to solve for example using convex solvers. One could therefore do iterative gradient updates on the upper level while directly solving the lower-level at each iteration:

$$\theta' = \theta - \alpha \nabla_\theta \mathbb{E}_{x \sim \mu_{\mathbf{X}}} \left[ \mathbb{KL}(\hat{\pi}^\star_{\mathbf{X}, \mathbf{Y}_\theta} \| \pi_\theta)[x] \right]$$
$$\text{s.t. } \hat{\pi}^\star_{\mathbf{X}, \mathbf{Y}_\theta} \in \arg\max_{\pi \in \Pi \cap \hat{\Phi}_+(\mathbf{X}, \mathbf{Y}_\theta)} \hat{\mathcal{J}}(\pi; \mathbf{X}, \mathbf{Y}_\theta)$$
$$(4)$$

where $\theta$ and $\theta'$ are the parameters of current and next iteration, respectively; and the projection step is also done with samples $(\mathbf{X}, \mathbf{Y}_\theta)$ of the current iteration.

Note that $\hat{\pi}^\star$ is indirectly a function of $\theta$ through the samples $(\mathbf{X}, \mathbf{Y}_\theta)$. The gradient $\nabla_\theta \mathbb{KL}(\hat{\pi}^\star_{\mathbf{X}, \mathbf{Y}_\theta} \| \pi_\theta)$ hence requires differentiation through $\hat{\pi}^\star_{\mathbf{X}, \mathbf{Y}_\theta}$, meaning differentiate through an $\arg\max$ operator, which can in principle be achieved through implicit differentiation (Amos & Kolter, 2017; Lorraine et al., 2020). However, to avoid additional implementation and computation overhead, instead we use the semi-gradient $\nabla_\theta \mathbb{KL}(\text{SG}\{\hat{\pi}^\star_{\mathbf{X}, \mathbf{Y}_\theta}\} \| \pi_\theta)$, where $\text{SG}\{\cdot\}$ is a stop gradient operator. This stop-gradient trick is quite common in many ML applications (Sutton, 2018; Foerster et al., 2018; Chen & He, 2021), leading to the update:

$$\theta' = \theta - \alpha \nabla_\theta \mathbb{E}_{x \sim \mu_{\mathbf{X}}} \big[ \underbrace{\mathbb{KL}(\text{SG}\{\hat{\pi}^\star_{\mathbf{X}, \mathbf{Y}_\theta}\} \| \pi_\theta)[x]}_{\text{approximate projection / weighted SFT}} \big]$$
$$\text{s.t. } \underbrace{\hat{\pi}^\star_{\mathbf{X}, \mathbf{Y}_\theta} \in \arg\max_{\pi \in \Pi \cap \hat{\Phi}_+(\mathbf{X}, \mathbf{Y}_\theta)} \hat{\mathcal{J}}(\pi; \mathbf{X}, \mathbf{Y}_\theta)}_{\text{optimization for weight}}$$
$$(5)$$

The semi-gradient leads to weighted SFT update (via $\hat{\pi}^\star$) as $\nabla_\theta \mathbb{KL}(\text{SG}\{\hat{\pi}^\star\} \| \pi_\theta)[x_i] = -\sum_j \hat{\pi}^\star(y_{ij}|x_i) \frac{\partial}{\partial \theta} \log \pi_\theta(y_{ij}|x_i)$. This observation creates an extremely simple update with negligible implementation overhead.

**Discussion.** The update rule ended up aligning many iterative weighted SFT algorithms, such as RAFT (Dong et al., 2023) and RFT (Ouyang et al., 2022; Touvron et al., 2023). In hindsight, our algorithm is motivated by the observation that these extremely successful algorithms can be interpreted as projecting empirical solutions onto a parametric space. Consequently, it is reasonable to use the empirical estimate in Eq. (2), as RAFT and RFT have demonstrated

Table 1: SFT methods from our optimization point-of-view. (w)-$\mathbb{XE}$ denotes (weighted) cross-entropy loss. RM stands for reward model. Binary reward indicates correctness. $\hat{\pi}^\star$ are (unnormalized) weights for the subsequent $\mathbb{XE}$ update.

| **ALGO** | loss $\mathcal{L}$ | reward $r$ | feas. set $\Phi$ | acceptance or weight $\hat{\pi}^\star$ |
|---|---|---|---|---|
| SFT | $\mathbb{XE}$ | constant | $\Pi$ | constant |
| RFT | w-$\mathbb{XE}$ | binary | $\Pi$ | $\mathbb{1}\{r(x,y)=1\}$ |
| RAFT (Best-of-$N$) | w-$\mathbb{XE}$ | RM | $\Pi$ | $\mathbb{1}\{r(x_i,y)=r^\star\}$, $r^\star=\max_j r(x_i,y_{ij})$ |
| Ours | w-$\mathbb{XE}$ | Sec. 3 | Eq. (2) | Eq. (2) |

strong practical performance despite the inherent bias introduced by the non-linearity of these estimations. Since it is essentially generalizes SFT by re-weighting a sample pair $(x_i, y_{ij})$ by $\hat{\pi}^\star(y_{ij}|x_i)$, at each iteration based on the solution of an optimization problem $\hat{\pi}^\star$, we create a Table 1 to outline the corresponding interpretation for some iterative SFT methods, from this optimization point-of-view.

The optimization problems for RFT and RAFT are trivial, as they assign $\hat{\pi}^\star(y|x) = 1$ to accepted responses and to the response with the highest reward model score, respectively. Formulating these methods as optimization does not offer much advantages. However, this perspective provides flexibility for future work to extend our framework, allowing for feasible sets and weighting tailored to specific applications.

## 3. Practical Implementation

Yet, as we are working in an algorithm-agnostic fashion, we are now ready to select a specific RL algorithm, define its corresponding objective $\mathcal{J}$, and specify a reward function $r$.

**Reward function.** Since we are working on mathematical problem-solving, a ground-truth reward could be obtained through string matching (Cobbe et al., 2021; Hendrycks et al., 2021b) of the model's solution against the ground truth solution, yielding a binary reward function $r_{\text{match}} : \mathcal{X} \times \mathcal{Y} \to \{0,1\}$ that indicates correctness. On top of the binary reward, we define our reward $r_\Delta$ as the reward margin. To formally construct the margin, we first define the expected reward of a set $\mathcal{G}$ such that $\bar{r}_\pi(x, \mathcal{G}) := \mathbb{E}_{y\sim\pi}[r_{\text{match}}(x, y) \mid y \in \mathcal{G}]$. We then define the reward margin $r_\Delta$ as the reward advantage of a group $\mathcal{G}$ against all other groups $\mathcal{Y} \setminus \mathcal{G}$:

$$r_\Delta(x, y \in \mathcal{G}) := \bar{r}_\pi(x, \mathcal{G}) - \bar{r}_\pi(x, \mathcal{Y} \setminus \mathcal{G}) \quad (6)$$

We have $r_\Delta(x, y \in \mathcal{G}_+) = \bar{r}_\pi(x, \mathcal{G}_+) - \bar{r}_\pi(x, \mathcal{G}_\circ)$, in our case, and similarly for $\mathcal{G}_\circ$. $r_\Delta$ may appear odd for not counting the correctness of individual $y$. This is handled subsequently.

**RL objective.** For the learning algorithm, our choice is Constraint Generative Policy Optimization (CGPO) (Xu et al., 2024b), which is designed for multi-objective constrained

optimization of LLMs. The choice is driven by implementation considerations: CGPO's modular constraint-handling design makes it straightforward to incorporate additional constraints, such as the group density constraint in our case. CGPO in a nutshell can be viewed as a generalized Best-of-$N$ (BoN), though depending on the specific CGPO settings. It operates by defining a feasible set $\Xi$ over the sample space $\mathcal{X} \times \mathcal{Y}$, which are designed to capture constraint adherence for, e.g. correctness, factuality, and safety (Xu et al., 2024b). In short, $(x, y) \notin \Xi$ will be rejected. In the context of math reasoning, the objective of CGPO can be summarized as:

$$\max_\pi \mathbb{E}_x \mathbb{E}_{y\sim\pi}[r(x, y)] \quad \text{s.t. } \textstyle\sum_y[\pi(y|x)\mathbb{1}_{\{y\in\Xi_x\}}] \geq 1 \ \forall x$$

$$\Xi_x := \underbrace{\{y : r_{\text{match}}(x, y) = 1\}}_{\text{correctness}} \cap \underbrace{\{y : \hat{\mathbb{KL}}(y; x, \pi_{\text{ref}}) \leq \mathbb{KL}_{\text{max}}\}}_{\text{empirical KL}}$$

(7)

This objective essentially optimizes reward over the feasible sets $\Xi_x$ such that feasible response $y$ is correct and within an KL range of $\mathbb{KL}_{\text{max}}$, where the KL constraint is measured using point estimate of the (forward) KL defined as $\hat{\mathbb{KL}}(y; x, \pi_{\text{ref}}) := \log \pi(y|x) - \log \pi_{\text{ref}}(y|x)$.

**Resulted update (IBPO w/ OPT$_{\text{IuB}}$).** Recap our update is defined as $\theta' = \theta - \alpha \nabla_\theta \mathbb{E}_{x\sim\mu_{\mathbf{x}}} \left[ \mathbb{KL}(\text{SG}\{\hat{\pi}^\star_{\mathbf{X},\mathbf{Y}_\theta}\} \| \pi_\theta)[x] \right]$ s.t. $\hat{\pi}^\star_{\mathbf{X},\mathbf{Y}_\theta} \in \arg\max_{\pi\in\Pi\cap\hat{\Phi}_+(\mathbf{X},\mathbf{Y}_\theta)} \hat{\mathcal{J}}(\pi; \mathbf{X}, \mathbf{Y}_\theta)$. And we have now defined the RL objective $\mathcal{J}$ and the margin reward function $r_\Delta$. We are now ready to put everything together:

$$\hat{\pi}^\star_{\mathbf{X},\mathbf{Y}_\theta} \in \arg\max_\pi \hat{\mathcal{J}}_\Delta(\pi; \mathbf{X}, \mathbf{Y}_\theta) :=$$
$$\textstyle\frac{1}{nm} \sum_i^n \sum_j^m [\pi(y_{ij}|x_i) r_\Delta(x_i, y_{ij})]$$
$$\text{s.t.} \quad \pi \in \Pi \cap \hat{\Phi}_+(\mathbf{X}, \mathbf{Y}_\theta) \text{ and} \quad (8)$$
$$\underbrace{\textstyle\sum_{y\in\mathbf{Y}_\theta}[\pi(y|x)\mathbb{1}_{\{y\in\Xi_x\}}] \geq 1 \ \ \forall x \in \mathbf{X}}$$

shorthanded as OPT$_{\text{IuB}}$ (inference under budget)

In addition, as CGPO is a generalization of BoN (with tie breaking randomly), $\hat{\pi}^\star$ will be a pure strategy instead of stochastic one, meaning at most one $y$ will be accepted for each $x$, for subsequent projection (SFT).

**Intuition.** Behind this update our intuition can be verbally interpreted as: If a prompt $x$ is hard, the margin of extended responses $r_\Delta(x, y \in \mathcal{G}_+) = \bar{r}_\pi(x, \mathcal{G}_+) - \bar{r}_\pi(x, \mathcal{G}_\circ)$ is likely to be large for $y \in \mathcal{G}_+$. Therefore $\hat{\pi}^\star$ will more likely to assign positive weight to an extended response $y \in \mathcal{G}_+$ so that the objective receive larger margin reward. In contrast, if a query $x$ is simple, $r_\Delta(x, y \in \mathcal{G}_+)$ is likely to be small. Hence $\hat{\pi}^\star$ will possibly assign positive weight to regular responses $y \in \mathcal{G}_\circ$, so that one could save some density budget for harder queries. See a concrete example in Table 2.

**Reward models.** Note that the original CGPO implementation has reward models for BoN ranking. We however intentionally excludes reward models, in our OPT$_{\text{IuB}}$ formulation as shown in Eq. (8), to highlight our methodological contributions, by decoupling reward modeling efforts.

Table 2: An example of $\text{OPT}_{\text{IuB}}$ compared to $\text{OPT}_{\text{cgpo}}$-Eq. (7)-with tie broken randomly, resulting in potentially non-unique $\hat{\pi}^\star$. This table shows several such $\hat{\pi}^\star$ solutions but not all. KL is omitted for brevity. Given the rewards below we have $\bar{r}(x_1, \mathcal{G}_\circ) = \bar{r}(x_1, \mathcal{G}_+) = \bar{r}(x_2, \mathcal{G}_+) = 1$ and $\bar{r}(x_2, \mathcal{G}_\circ) = 0.5$. Suppose the cap $q_+ = 0.5$, allowing at most 50% of accepted responses are extended $y \in \mathcal{G}_+$. For a solution matrix $\hat{\pi}^\star$, 1 and 0 represent accepted and rejected response, respectively.

| prompt | $x_1$ (easy) | $x_2$ (hard) |
|---|---|---|
| responses | $y_{11}, y_{12} \in \mathcal{G}_\circ$ $y_{13}, y_{14} \in \mathcal{G}_+$ | $y_{21}, y_{22} \in \mathcal{G}_\circ$ $y_{23}, y_{24} \in \mathcal{G}_+$ |
| $r(x, y)$ | $\begin{pmatrix} 1 & 1 \\ 1 & 1 \end{pmatrix}$ | $\begin{pmatrix} 1 & 0 \\ 1 & 1 \end{pmatrix}$ |
| $r_\Delta(x, y)$ | $\begin{pmatrix} 0 & 0 \\ 0 & 0 \end{pmatrix}$ | $\begin{pmatrix} -\frac{1}{2} & -\frac{1}{2} \\ \frac{1}{2} & \frac{1}{2} \end{pmatrix}$ |
| $\hat{\pi}_1^\star = \text{OPT}_{\text{cgpo}}{}^\dagger$ | $\begin{pmatrix} 0 & 0 \\ 1 & 0 \end{pmatrix}$ | $\begin{pmatrix} 1 & 0 \\ 0 & 0 \end{pmatrix}$ |
| $\hat{\pi}_2^\star = \text{OPT}_{\text{cgpo}}{}^\ddagger$ | $\begin{pmatrix} 0 & 0 \\ 0 & 1 \end{pmatrix}$ | $\begin{pmatrix} 0 & 0 \\ 1 & 0 \end{pmatrix}$ |
| $\hat{\pi}^\star = \text{OPT}_{\text{IuB}}{}^\sharp$ | $\begin{pmatrix} 1 & 0 \\ 0 & 0 \end{pmatrix}$ | $\begin{pmatrix} 0 & 0 \\ 1 & 0 \end{pmatrix}$ |

$\dagger$ CGPO case 1: for $x_2$ (hard), $y_{21} \in \mathcal{G}_\circ$ is accepted even though $\mathcal{G}_+$ has higher expected reward $\bar{r}(x_2, \mathcal{G}_+)$; $\ddagger$ case 2: $y_{14}$ and $y_{23}$ are both accepted, which exceeds the density budget $q_+ = 0.5$.
$\sharp$ Ours: accepts $y_{11}$ and $y_{23}$ to maximize margin, $r_\Delta(x_2, y_{23}) = 0.5$, while adhering to the density cap $q_+ = 0.5$.

Nonetheless, using reward models remains possible. To avoid introducing additional notation, we elaborate verbally: $\text{OPT}_{\text{IuB}}$ essentially selects either group $\mathcal{G}_+$ or $\mathcal{G}_\circ$ for a query $x$. Within a group $\mathcal{G}$, all responses receive the same reward $r_\Delta(x, \mathcal{G})$, leaving it possible to further rank responses within each group using reward models.

**Implementation & solvers.** A pseudo-code of our IBPO with $\text{OPT}_{\text{IuB}}$ is listed in Algorithm 1. The essential change is to replace the constrained reward ranking $\text{OPT}_{\text{CGPO}}$ with a general optimization problem, in our case the margin maximization under budget denoted as $\text{OPT}_{\text{IuB}}$. The $\text{OPT}_{\text{IuB}}$ problem is a (integer) linear programming problem that could be solved by off-the-shelf solvers, such as CPLEX (Cplex, 2009), Gurobi (Gurobi Optimization, LLC, 2024), or SciPy (Virtanen et al., 2020) which is our choice.

## 4. Naïve Construction of $\mathcal{G}_+$

Yet we work on abstract groups $\mathcal{G}_\circ$ and $\mathcal{G}_+$. In this section, we present the details of our constructions of extended length responses, i.e. the group $\mathcal{G}_+$. However developing long reasoning models is beyond the scope of this work, as our focus is on the constrained optimization of LLMs. Our constructions are for demonstrative purpose only.

---

**Algorithm 1** Inference Budget-Constrained PO (IBPO)

**Require:** prompt set $\mathbb{D}$, batch size $n$, number of responses $m$, init policy $\pi_0 = \pi_{\text{ref}}$, num of iters $T$, budgets $q_+$
1: **for** $t = 1, \ldots, T$ **do**
2:     prompt sampling: $\mathbf{X}^n \sim \mu_{\mathbb{D}}$
3:     response generation: $\mathbf{Y}_\theta^{n \times m} \sim \pi_{\theta_t}(\mathbf{X})$
4:     evaluate correctness: $\mathbf{R}_c^{n \times m} = r_{\text{match}}(\mathbf{X}, \mathbf{Y})$
5:     empirical KL: $\hat{\mathbf{KL}}^{n \times m} = \hat{\mathbb{KL}}(\mathbf{Y}; \mathbf{X}, \pi_{\text{ref}})$
6:     **if** CGPO (i.e. w/o IuB) **then**
7:         solve BoN (7): $\hat{\pi}_{\mathbf{X}, \mathbf{Y}_\theta}^\star \in \text{OPT}_{\text{cgpo}}(\mathbf{X}, \mathbf{Y}_\theta, \mathbf{R}_c, \hat{\mathbf{KL}})$
8:     **end if**
9:     **if** IBPO (i.e. w/ IuB) **then**
10:        solve Eq. (8): $\hat{\pi}_{\mathbf{X}, \mathbf{Y}_\theta}^\star \in \text{OPT}_{\text{IUB}}(\mathbf{X}, \mathbf{Y}_\theta, \mathbf{R}_c, \hat{\mathbf{KL}}, q_+)$
11:     **end if**
12:     grad step: $-\sum_i \sum_j \hat{\pi}_{\mathbf{X}, \mathbf{Y}_\theta}^\star(y_{ij} | x_i) \frac{\partial}{\partial \theta} \log \pi_\theta(y_{ij} | x_i) \big|_{\theta = \theta_t}$
13: **end for**

---

Due to the intricate details and the space limit, we defer the full version of this section to Appendix B, including details of prompting, dataset construction, training pipelines, etc.

Figure 1 provides examples of our constructions, which should be visually telling. In a nutshell, we use the step CoT (SCoT) format from Dubey et al. (2024) as $\mathcal{G}_\circ$, and construct a sequential voting (SV) response as $\mathcal{G}_+$. SV serves as a unimodal baseline to demonstrate the performance of our construction. Adaptive SV (ASV) generates a mixture of SCoT and SV responses, allowing us to optimize its ability to adaptively choose between the two. For further details about the constructions, refer to Figure 1 and Appendix B.

**Notations.**, we use LLaMA to refer to instruction-tuned LLaMA 3.1 8B (Dubey et al., 2024). ASV-SFT-$\alpha$ denotes ASV models that are supervise fine-tuned with a coefficient $\alpha$ (see Appendix B). ASV-IuB-$q_+$ refers to ASV models optimized by our Algorithm 1 with a budget constraint $q_+$.

## 5. Evaluation of IBPO w/ $\text{OPT}_{\text{IuB}}$

In this section, we show that: (i) SV is a valid construction with reasonable absolute improvement (Table 3) and scales as good as majority vote (MV, aka self-consistency (Wang et al., 2022)) (Figure 2a & 2b); (ii) ASV-SFT does not achieve good efficiency (Figure 2a & 2b) as SFT does not optimize efficiency as an optimization objective; (iii) ASV-IuB optimized by IBPO achieves better efficiency (Figure 2a & 2b), adherence to constraints (Figure 2c), and allocation of inference to harder quires (Figure 2e & Figure 2f).

### 5.1. Absolute Improvement (Table 3)

**"Baselines".** To put SV/ASV in comparison with other baselines in literature, we gather several baselines from the self-correction literature as essentially these methods in-

**SV Prompt (Simplified)**

You are asked to give at most eight diverse solutions in different way, without referencing to the previous trials. If a solution occurs three times, it is considered as a consensus and will be used as the final answer.

**ASV Prompt (Simplified)**

For medium and hard level problems, you are asked to give at most eight diverse solutions in different way, without referencing to the previous trials.
For easy level problems, you are allowed only one attempt, which will be considered your final answer.

**Voting Response $\mathcal{G}_+$ (SV, ASV Case 1)**

[TRIAL] ## Step 1: ... steps omitted
The final answer is: $\boxed{A1}$ . [/TRIAL]
[TRIAL] ## Step 1: ... steps omitted
The final answer is: $\boxed{A2}$ . [/TRIAL]
[TRIAL] ## Step 1: ... steps omitted
The final answer is: $\boxed{A1}$ . [/TRIAL]
[TRIAL] ## Step 1: ... steps omitted
The final answer is: $\boxed{A1}$ . [/TRIAL]
The answer $\boxed{A1}$ has occurred three times, and is considered as a consensus.
The final answer is $\boxed{A1}$ . I hope it is correct.

**Non-Voting Response $\mathcal{G}_\circ$ (ASV Case 2)**

[TRIAL] ## Step 1: ... steps omitted
The final answer is: $\boxed{A1}$ . [/TRIAL]
Terminated due to difficulty level.

Figure 1: Example of prompts and responses. For $\mathcal{G}_\circ$, we use standard step CoT (SCoT). To construct $\mathcal{G}_+$, the model is prompted to sequentially generate up to 8 trials encapsulated within the special tokens [TRIAL] and [/TRIAL], followed by a consensus answer. The SV prompt instructs the model to output only voting responses, allowing us to later evaluate its performance and show that it is a reasonable construction. The ASV prompt asks the model to decide whether to output a voting or non-voting response, facilitating our further optimization of budget allocation in Section 5.

crease inference length, though not extensively as Jaech et al. (2024). The results in Table 3 are mainly gathered from Qu et al. (2024); Kumar et al. (2024). The self-correction baselines often admit a multi-turn structure, similarly to the multi-trial construction of SV. We therefore include a column of inference cost measured by number of turns/trials for a rough comparison. The SFT comparators we include are (reproduced) Self-Refine (Madaan et al., 2024), STaR (Ze-

Table 3: Comparison of approaches on their improvements versus base models. $^\dagger$, $^\ddagger$, $^{\dagger\dagger}$ indicate results duplicated from Qu et al. (2024); Yan et al. (2024); Kumar et al. (2024), respectively. $^*$ indicates our methods/constructions and the improvements are relative to LLaMA model.

| approach | pass@1 | improv. | turns/trials per response |
|---|---|---|---|
| SFT/Prompting-based | | | |
| **SV-SFT**$^*$ | | | |
| LLaMA | 56.8 | 5.54 | 5.67x |
| **ASV-SFT-1**$^*$ | | | |
| LLaMA | 55.6 | 4.43 | 5.74x |
| **SFT-RISE**$^\dagger$ | | | |
| setting 1 (table 1 of $\dagger$) | 5.5 | -0.3 | 5x |
| setting 2 (table 1 of $\dagger$) | 5.0 | 0.0 | 5x |
| **SFT-SCoRe**$^{\dagger\dagger}$ | | | |
| setting 1 (table 1 of $\dagger\dagger$) | 54.2 | 1.8 | 2x |
| setting 2 (table 1 of $\dagger\dagger$) | 55.0 | 0.0 | 2x |
| **RISE**$^\dagger$ | | | |
| LLaMA2 Base | 1.4 | -0.5 | 5x |
| + boosting | 5.5 | 0.0 | 5x |
| **S³C**$^\ddagger$ | | | |
| LLaMA3-8B | 33.14 | 2.56 | |
| Mistral-7B | 25.48 | 1.44 | not studied |
| DeepSeek-Math-Base-7B | 41.40 | 3.18 | |
| Qwen2-Math-7B | 51.76 | 0.44 | |
| **Self-Refine**$^\dagger$ | | | |
| Base | 1.9 | 0.0 | 3x |
| GPT-3.5 | 36.5 | -3.2 | 3x |
| Mistral-7B | 7.1 | -0.4 | 3x |
| Eurus-7B-SFT | 9.0 | -3.3 | 3x |
| **STaR**$^{\dagger\dagger}$ | | | |
| setting 1 (table 1 of $\dagger\dagger$) | 54.0 | 0.4 | 2x |
| setting 2 (table 1 of $\dagger\dagger$) | 41.2 | -14.2 | 2x |
| Online Iterative/RL | | | |
| **ASV-IuB-$q_+$**$^*$ | | | |
| $q_+ = 25\%$ | 54.2 | 2.94 | 2.24x |
| $q_+ = 50\%$ | 55.4 | 4.14 | 2.16x |
| $q_+ = 75\%$ | 57.0 | 5.74 | 4.32x |
| **RISE**$^\dagger$ | | | |
| + Iteration 1 | 9.7 | 3.4 | 5x |
| + Iteration 2 | 10.4 | 4.6 | 5x |
| **SCoRe**$^{\dagger\dagger}$ | | | |
| Gemini 1.5 Flash | 64.4 | 4.4 | 2x |
| + more turns (fig. 8$^{\dagger\dagger}$) | ≈66 | ≈6 | 5x-10x |

likman et al., 2022), S³C (Yan et al., 2024), SFT-RI and SFT-SCoRe, where SFT-RI and SFT-SCoRe are SFT comparators implemented in Recursive Introspection (RI) (Qu et al., 2024) and SCoRe (Kumar et al., 2024) respectively.

**ASV experiments.** SV-SFT follows our training pipeline of Exp 1.2 in Table 7 (Appendix B) and is an voting only baseline. ASV-SFT-$\alpha$ follows our Exp 2.1 setup and is an adaptive baseline, meaning model decide whether to vote or

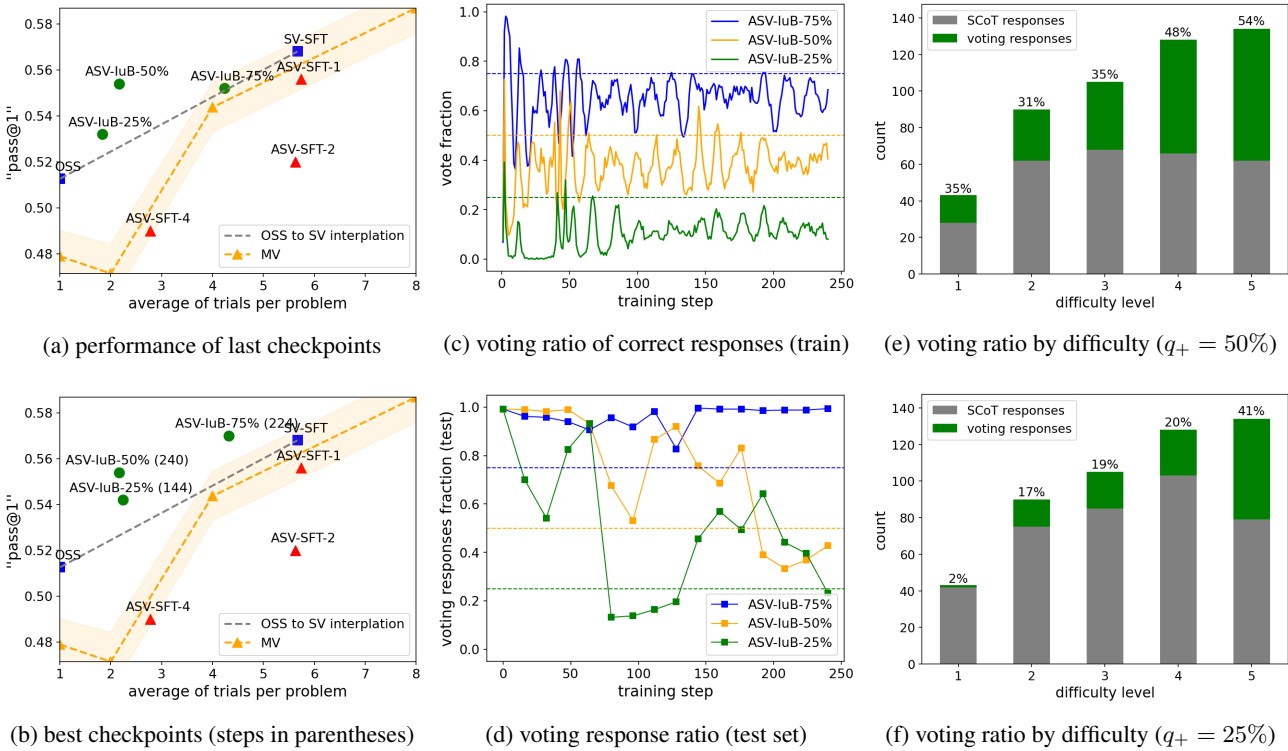

(a) performance of last checkpoints   (c) voting ratio of correct responses (train)   (e) voting ratio by difficulty ($q_+ = 50\%$)

(b) best checkpoints (steps in parentheses)   (d) voting response ratio (test set)   (f) voting ratio by difficulty ($q_+ = 25\%$)

Figure 2: Column 1: Comparison of pass@1 (maj@$N$ for MV) against the average number of trials per response (x-axis). OSS refers to LLaMA. The interpolation between OSS and SV-SFT (aligning with MV) serves as a hypothetical efficiency boundary. ASV-SFT shows lower efficiency relative to this boundary, whereas ASV-IuB consistently achieves better efficiency (i.e. above the boundary). Column 2: Voting response ratio versus training steps. Dashed line denotes the budget $q_+$. On the training set, IuB formulation follows the budget constraints almost exactly. Due to distribution shift, the constraint on testing set is not entirely exact, but still it is noticeable that the voting ratio follows the order of $75\% \succ 50\% \succ 25\%$. Column 3: The IuB formulation enables the model to dynamically allocate voting budget to harder problems.

not for a query. We report ASV-SFT-1 only as it is empirically the best ASV-SFT-$\alpha$ baseline, though still fall short in terms of efficiency. The ASV-IuB-$q_+$ experiments, initialized from ASV-SFT-1, are our RL-tuned adaptive models, optimized by IBPO with $\text{OPT}_{\text{IuB}}$ as inner optimization.

**Observations.** We would like to first emphasize that the comparison in Table 3 is not intended to demonstrate that SV outperforms SFT-based self-correction or that our ASV-IuB surpasses RL-based self-corrections. These efforts are orthogonal, as our focus is on constrained optimization. As observed, our SFT constructions—SV-SFT and ASV-SFT-1 achieve a clear improvement in pass@1 with high inference costs (5+ times the number of trials). The ASV-IuB-$q_+$ formulation, particularly with $q_+ = \{50\%, 75\%\}$, shows significant improvement while reducing costs by $4.14\%$ at $2.16\times$ and $5.74\%$ at $4.32\times$. This performance is on par with SCoRe, a state-of-the-art RL-based self-correction method. Note that the performance of ASV-IuB-$q_+$ is reported using the best checkpoints. Results from the last checkpoint are shown in Figure 2a. Additionally, training curves are presented in Appendix E, which shows consistent

improvements. As a somewhat tangential yet potentially intriguing observation, it is evident that prompting-based and SFT-based methods struggle with both absolute improvement (Table 3) and efficiency (Figure 2), supporting the conjecture that SFT alone does not enable self-correction capabilities (Huang et al., 2023; Kumar et al., 2024). This observation is also partially supported by concurrent work (DeepSeek-AI et al., 2025), which suggests that such self-correction emerges automatically during RL rather than manually created by prompting or SFT.

### 5.2. Efficiency, Budget Adherence & Allocation

Table 3 shows that our ASV-IuB-$q+$ models achieve results comparable to RL-based self-correction models, simply by inference budget management. In this subsection, we extend our discussion on: (i) performance-budget efficiency, (ii) constraint satisfaction, and (iii) inference budget allocation.

**Performance-budget efficiency.** In Figure 2a and 2b, we visually assess the performance-budget efficiency, compared to a hypothetical efficiency boundary. This boundary is

an interpolation between OSS LLaMA model and SV-SFT. It is reasonable to consider it as a hypothetical boundary for two reasons: (i) OSS and SV-SFT are two extremes of ASV-IuB-$q_+$, corresponding to the cases of $q_+ = 0$ and $q_+ = 1$ respectively; and (ii) this interpolation achieves an increase comparable to MV, if not slightly better. The SFT version of ASV is generally much worse than the boundary, as SFT alone does not optimize resource allocation as a mathematical objective. For ASV-IuB-$q_+$ optimized by IBPO, we report both the last and best checkpoint results in Figure 2a and Figure 2b, respectively. Our formulation achieves, in general, better performance-budget efficiency, except $q_+ = 75\%$ in Figure 2a. We will extend our discussion on this unsuccessful case in subsequent paragraphs.

**Constraint satisfaction.** We then evaluate how effectively the constraints are enforced. In Figure 2c, the budget constraints are successfully maintained during training for $q_+ = \{25\%, 50\%, 75\%\}$. Due to distribution shifts, exact adherence to these constraints is not expected on the test set. Nevertheless, Figure 2d demonstrates that constraints are still upheld at the end of training for $q_+ = \{25\%, 50\%\}$, and the ratio of voting responses follows the order $75\% \succ 50\% \succ 25\%$. (Figure 2c illustrates that our model meets the budget constraints for the set of correct responses. The constraints also hold for all responses, see Appendix E.)

**Difficulty-adaptive allocation.** We have show improved efficiency and adherence to constraints, we would like to further validate the intuition of our design: that more challenging problems may require longer reasoning steps, whereas simpler problems can be resolved with just SCoT responses. Ideally, the model should allocate more voting responses to problems with higher difficulty levels. To this end, we use the difficulty levels from the metadata of Hendrycks MATH and plot the ratio of voting responses for each difficulty level. Figure 2f and 2e illustrates that for both $q_+ = \{25\%, 50\%\}$, more challenging problems, such as those at levels 4 and 5, receive a higher allocation of budgets. This allocation pattern is particularly evident for the case of $q_+ = 25\%$, where only 2% of level 1 problems receive voting responses.

**Discussion on the unsuccessful case.** There is one observed unsuccessful case in Figure 2a: the last checkpoint of ASV-IuB-75%, which falls on the hypothetical boundary rather than above it. This outcome is arguably expected, as observed in Figure 2d, where ASV-IuB-75% outputs almost exclusively voting responses at the end of training. As a result, this model is not adaptively allocating resources, and thus no improvement in efficiency is anticipated. This unsuccessful case is hence caused by the distribution shift between the training set and the testing set. It is possible that scaling the training set—given that our training set $\mathbb{D}_{\mathrm{RL}}$ contains approximately 10k prompts, which is relatively small—will make the testing set more likely to be in-distribution and thereby alleviate the distribution shift issue.

## 6. Conclusion & Discussions

We derived a constrained policy optimization framework, IBPO, from an optimization perspective, resulting in a simple weighted SFT update that resembles successful iterative SFT algorithms such as RFT and RAFT. In each iteration, the optimal weight is obtained by solving an (integer) linear program. The practical implementation of IBPO is build on top of CGPO, and is evaluated on a math reasoning task with inference budget constraints. Empirical evaluations show that our framework enables the model to adhere to constraints and dynamically allocate the inference budget.

**Batch optimization & solver time.** Since we solve an optimization problem per iteration (i.e. per mini-batch), limited computational resources can result in smaller sample sizes for the inner optimization problem, leading to larger variance. This issue can be mitigated through "sample accumulation", accumulating samples across multiple consecutive steps, similar to gradient accumulation practice in LLMs. A pseudo-code for sample accumulation can be found in Appendix H. In addition, though integer linear programming is NP-hard (Vazirani, 1997), the number of variables in our batch-level optimization is typically small, resulting in minimal computational overhead. Refer to the wall-time plot for the SciPy solver in Appendix H for details.

**Broader applications.** Our framework has only been evaluated with inference in math, the resource allocation problem however has far-reaching implications within the ML community. As a result, our framework can be potentially extended to further LLM applications. For example, a promising application is statistical parity (Zemel et al., 2013), aka group fairness. In this context, one could consider attributing responses to their respective social groups, and cap the density of responses that correspond to socially privileged groups, to encourage more inclusive and equitable responses across different demographics. Another potential application is the balanced expert activation in mixture of experts (Jacobs et al., 1991; Shazeer et al., 2017; Lepikhin et al., 2020) systems, which is sometimes achieved by adding an auxiliary balancing loss (Wei et al., 2024). Alternatively, this balance can be possibly achieved by enforcing a minimal total activation density for each expert. This may help to prevent over-reliance on a subset of experts, and hence enhancing the overall robustness and efficiency. We leave the exploration of broader applications and their implementations as future directions.

### Impact Statement

This paper presents work whose goal is to advance the field LLMs. There are many potential societal consequences of our work, none of which we feel must be highlighted here.

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

## A. Motivating Example

Figure 3 is an example that advanced reasoning model spent more than enough time on a trivial problem.

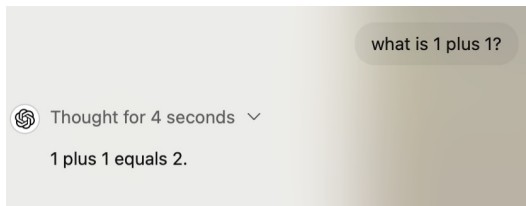

Figure 3: A long reasoning-chain model spent more than enough inference time on a trivial problem.

## B. Full Version of Section 4: Acronyms, Naïve Construction of $\mathcal{G}_+$ & Training Pipelines

Yet we work on abstract groups $\mathcal{G}_\circ$ and $\mathcal{G}_+$. In this section, we present the details of our constructions of extended length responses, i.e. the extended group $\mathcal{G}_+$. However developing long reasoning models is beyond the scope of this work, as our focus is on the constrained optimization of LLMs. Our constructions are for demonstrative purpose only. Due to the intricate details involved in prompts, datasets, and training pipelines, this section may appear somewhat dense. To make it more approachable, we have structured our writing in a way that readers can, if they wish, focus on the broader ideas without delving deeply into the specifics of constructions. A TL;DR version of this section is provided below.

**TL;DR.** We construct two types of illustrative extended responses: Sequential Voting (SV) and Adaptive Sequential Voting (ASV). Figure 4 visually explains how these constructions are implemented. The goal of the SV is to establish a baseline that generates only responses in $\mathcal{G}_+$, thereby serving as an uni-modal comparator. SV scales roughly as well as vanilla majority voting (MV), aka self-consistency (Wang et al., 2022). In contrast, ASV outputs a mixture of responses of $y \in \mathcal{G}_\circ$ and $y \in \mathcal{G}_+$. This allows the model to adaptively decide which type of response to produce based on the query. The goal of ASV is to further enable IBPO optimization, as IBPO implicitly assumes the model generates both regular and extended-length responses. In Section 5, we show that ASV optimized by IBPO, achieves better allocation of the inference budget.

### B.1. Construction of Sequential Vote

**Acronyms.** For clarity, we explicitly define key terms to hopefully resolve any potential ambiguities. *Response*: A response refers to a sequence generated until a terminal token is encountered. For precision, we sometimes refer to these as voting responses or SCoT responses, as illustrated in Figure 4, after introducing our sequential voting baselines. *Trial*: A trial denotes a solution instance, which is demarcated by the special tokens [TRIAL] ... [/TRIAL], as shown in the voting response example in Figure 4. While a voting response contains multiple trials, a SCoT response or a non-voting response contains exactly one trial, as also depicted in Figure 4.

For the description of training and testing details, we use LLaMA and LLaMA-b to denote the instruction-tuned and base versions of the LLaMA 3.1 8B models (Dubey et al., 2024), respectively. MATH refers specifically to the training split of the Hendrycks MATH dataset (Hendrycks et al., 2021b), while the 500-sample subset of the testing split is referred to as MATH500 (Lightman et al., 2023). SDPO stands for step-DPO (Lai et al., 2024) dataset, a curated step-annotated dataset from which we retain only the prompts and positive responses (ground truth solutions), excluding any step signals (see Appendix G for details). It is important to note that while we leverage their curated dataset, the original SDPO method is not relevant to this work. The SDPO dataset was chosen because its ground truth responses follow the SCoT format of LLaMA responses, making it convenient to run supervised fine-tuning (SFT) mixed with LLaMA samples.

**Construction details.** To be more specific, the naïve sequential voting baselines, as the "expensive" group $\mathcal{G}_+$, have increased inference costs by simply sequentially output multiple trials and find the sequential majority vote until stop condition met. In particular, the early-stopping sequential voting (SV) baseline is created to show that such naïve baseline could achieve performance gain, on par with vanilla majority voting (MV). The adaptive sequential voting (ASV) allows model to output both SCoT response $y_\circ$ and sequential voting response $y_+$, allowing us to further conduct the budget controll experiments in Section 5.

- Early Stopping Sequential Vote (SV): For a single response, model are allowed to output at most 8 trials, and conclude

---

**SCoT Prompt + Cond.**

Solve the following math problem efficiently and clearly:
- For simple problems (2 steps or fewer):
Provide a concise solution with minimal explanation.
- For complex problems (3 steps or more):
Use this step-by-step format:
## Step 1: [Concise description]
[Brief explanation and calculations]
## Step 2: [Concise description]
[Brief explanation and calculations]
...
Regardless of the approach, always conclude with:
Therefore, the final answer is: $\\boxed{answer}$. I hope it is correct.
Where [answer] is just the final number or expression that solves the problem.
[Condition Description]
Problem: {{ problem }}

---

**SCoT Response**

## Step 1: ... steps omitted
The final answer is: A1 . I hope it is correct.

---

**SV Cond.**

You are asked to give at most eight diverse solutions in different way, without referencing to the previous trials.
Each trial should be contained in a separate [TRIAL] trial solution [/TRIAL] block.
If a solution occurs three times, it is considered as a consensus and will be used as the final answer.
If there is no consensus, use the solution from the most plausible trial.

---

**ASV Cond.**

For medium and hard level problems, you are asked to give at most eight diverse solutions in different way, without referencing to the previous trials.
Each trial should be contained in a separate [TRIAL] trial solution [/TRIAL] block.
If a solution occurs three times, it is considered as a consensus and will be used as the final answer.
If there is no consensus, use the solution from the most plausible trial.
For easy level problems, you are allowed only one attempt, which will be considered your final answer.

---

**Voting Response (SV, ASV Case 1)**

[TRIAL] ## Step 1: ... steps omitted
The final answer is: A1 . [/TRIAL]
[TRIAL] ## Step 1: ... steps omitted
The final answer is: A2 . [/TRIAL]
[TRIAL] ## Step 1: ... steps omitted
The final answer is: A1 . [/TRIAL]
[TRIAL] ## Step 1: ... steps omitted
The final answer is: A1 . [/TRIAL]
The answer A1 has occurred three times, and is considered as a consensus.
The final answer is A1 . I hope it is correct.

---

**Non-Voting Response (ASV Case 2)**

[TRIAL] ## Step 1: ... steps omitted
The final answer is: A1 . [/TRIAL]
Terminated due to difficulty level.

---

Figure 4: Prompt templates. For $\mathcal{G}_\circ$, we use the SCoT prompt without including an additional [condition description] and generate a standard SCoT response. For $\mathcal{G}_+$, we simply insert the corresponding condition into the [condition description] placeholder to create a new prompt. In the case of SV, the SV condition is integrated into the prompt, and the model is asked to perform repeated trials to reach a consensus. The ASV prompt instructs the model to output either a voting response (case 1) or a non-voting response (case 2), with the decision made by the model itself.

with majority of trials. In addition to the terminal condition of maximum 8 trials, model will early stop if an answer appears 3 times and this answer will be considered as the majority answer.

- Adaptive Sequential Vote (ASV): Model is allowed to choose either vote (case 1, i.e. $y_+$) or not (case 2, i.e. $y_\circ$). This baseline is created to further allow model-driven resource allocation. We later, in Section 5, show that one could optimize the ability of resource allocation with our IBPO.

**Dataset.** We define our construction of dataset as a product of a problem set $\mathbb{Q}$, format template of question $\mathcal{T}_q$ and answer $\mathcal{T}_a$, and a response set $\mathbb{A}$, subjected to some filtration $\mathcal{F}$. Formally, a dataset $\mathbb{D}$ is defined as: $\mathbb{D} := (\mathcal{F} \circ \mathcal{T}_q \circ \mathcal{T}_a)(\mathbb{Q} \times \mathbb{A})$, where $\mathcal{F}$ removes undesired question-answer pairs, such as incorrect responses when $\mathbb{A}$ are model generated; The templates $\mathcal{T}_q$ and $\mathcal{T}_a$ collectively transform each question-answer pair into a specific text format, as shown in Figure 4; Slightly abusing the notation, the Cartesian product $\mathbb{Q} \times \mathbb{A}$ is used to pair each response with its corresponding question, defined as: $\mathbb{Q} \times \mathbb{A} := \{(q_i, a_{ij}) : \forall i, j\}$.

Specifically, we summarize the datasets used in subsequent sections in Table 4. For instance, for SV training, we construct

Table 4: Summary of constructed datasets for different experimental purposes. $\mathbb{D}_{SV}$, $\mathbb{D}_{ESSV}$ and $\mathbb{D}_{ASV}$ uses the same set of prompts but different prompt/response templates. $\mathbb{Q}_{SDPO}$ and $\mathbb{A}_{SDPO}^{GOLDEN}$ are from Lai et al. (2024) with details deferred to Appendix G. ASV1 and ASV2 correspond to ASV case 1 and 2 (see Section B.1) respectively.

| Set | $\mathcal{T}_q$ | $\mathcal{T}_a$ | $\mathcal{F}$ | $\mathbb{Q}$ | $\mathbb{A}$ |
|---|---|---|---|---|---|
| $\mathbb{D}_{SV}$ | SV | SV | G | $\mathbb{Q}_{MATH}$ | $\mathbb{A}_{MATH}^{SAMPLE}$ |
| $\mathbb{D}_{ASV1}$ | ASV | ASV1 | | | |
| $\mathbb{D}_{ASV2}$ | ASV | ASV2 | - | $\mathbb{Q}_{SDPO}$ | $\mathbb{A}_{SDPO}^{GOLDEN}$ |
| $\mathbb{D}_{SCOT}$ | SCoT | SCoT | - | | |
| $\mathbb{D}_{RL}$ | ASV | - | - | $\mathbb{Q}_{SDPO}$ | $\emptyset$ |

Table 5: Stopping conditions.

| | stopping conditions |
|---|---|
| SV | (i) max 8 trials; (ii) if an answer occurs 3 times. |
| ASV | voting (case 1): (i) max 8 trials; (ii) if an answer occurs 3 times; non-voting (case 2): exact 1 trial. |

Table 6: Prompt and response sources.

| Set | Query-Response | Source |
|---|---|---|
| Training | $\mathbb{Q}_{MATH}$ & $\mathbb{A}_{MATH}^{SAMPLE}$
$\mathbb{Q}_{SDPO}$ & $\mathbb{A}_{SDPO}^{GOLDEN}$ | MATH & LLaMA samples (Lai et al., 2024) |
| Testing | $\mathbb{Q}_{MATH500}$ | MATH500 |

Table 7: Training pipelines. For Sec. B.2, we aim to create a demonstration experiment showing that with the same model (of roughly same math knowledge) SV can achieve reasonable performance-cost efficiency on par with MV. Sec. 5 further show we can optimize performance-cost efficiency through our IuB generalization of CGPO, where $\alpha$ in row 2.1 is a coefficient of $\mathbb{D}_{ASV2}$.

| Exp. | Sec. | Type | Init. Ckpt. | Dataset | Purpose |
|---|---|---|---|---|---|
| 1.1 | Sec. B.2 | SFT | LLaMA-b | $\mathbb{D}_{SV} \cup \mathbb{D}_{SCOT}$ | Allow model follow both SV and SCoT prompt. |
| 1.2 | Sec. 5 | SFT | LLaMA | | |
| 2.1 | Sec. 5 | SFT | LLaMA | $\mathbb{D}_{ASV1} \cup \alpha\mathbb{D}_{ASV2}$ | Follow the ASV instruction to let model decide vote or not. |
| 2.2 | | RL | Exp. 2.1 | $\mathbb{D}_{RL}$ | Optimize the capability of dynamic budget allocation. |

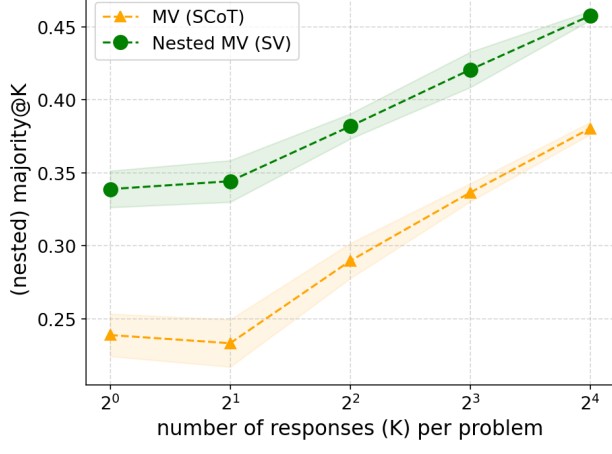

(a) (Nested) MV measured with number of responses

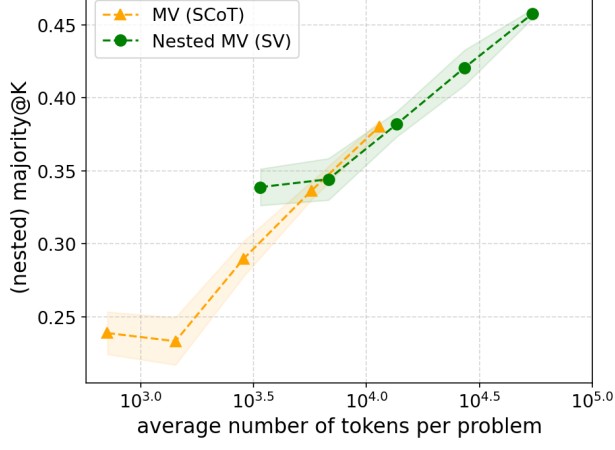

(b) (Nested) MV measured with number of tokens

Figure 5: SV tested on MATH500. MV stands for vanilla majority voting, aka self-consistency (Wang et al., 2022), with SCoT responses. Nested MV refers to majority voting with our (early-stopping) SV responses. It is "nested" as each SV response is already a voting, as shown in Figure 4. The SV method has "clear" gains when performance is measured with the number of responses. When measured with the number of tokens, SV aligns the performance-cost efficiency of vanilla MV. This is another indicator that one should worry about token efficiency when measuring reasoning performance.

$\mathbb{D}_{SV}$ with question set $\mathbb{Q}_{MATH}$ from Hendrycks et al. (2021b) and LLaMA generated responses $\mathbb{A}_{MATH}^{SAMPLE}$ using the SV templates

defined in Figure 4, subjected to data selection described in Appendix G.

**Training pipelines.** We summarize our training pipelines for our toy experiments in Section B.2 and our IuB experiments in Section 5. For instance, experiment 2.2 (Section 5) in Table 7 is the summary of our IBPO training, using dataset $\mathbb{D}_{\mathrm{RL}}$ as constructed in Table 4 and initialized from ASV models from experiment 2.2. Further details of each training can be found in Appendix G and F.

## B.2. Performance of Naïve Sequential Vote

We start by evaluating the performance of SV compared to vanilla MV, following the setup of Exp. 1.1 in Table 7. This toy experiment is designed to demonstrate: (i) SV scales approximately as well as MV, thereby qualifying it as an example of $\mathcal{G}_+$; and (ii) measuring performance based on the number of responses is inadequate, therefore we later measure performance relative to the number of tokens/trials in Section 5.

**Metrics.** The metrics we use are pass@$k$ and majority@$k$, both of which are widely used in the literature (Hendrycks et al., 2021b; Wang et al., 2022). We occasionally refer to pass rate as pass@1. In both metrics, $k$ specifically denotes the number of responses, regardless of the number of trials per response. Since our voting methods may involve multiple trials, we may use the average number of trials (as illustrated in Figure 2) on the x-axis for cost-aware comparisons. In addition, we evaluate "performance-cost" efficiency by comparing each method's scaling efficiency to that of MV, following the comparison in Snell et al. (2024).

As shown in Figure 5a, SV exhibits significant improvements—nearly 10%—in terms of majority@$k$ when performance is measured by the number of responses. However, this improvement is misleading since each SV response effectively consists of multiple SCoT responses. Therefore, measuring performance with taking length into consideration provides a more reasonable assessment. As demonstrated in Figure 5b, SV's scaling performance aligns with that of vanilla MV, supporting its role as a suitable construction example for $\mathcal{G}_+$.

## C. Batch Accumulation

This is a batch accumulation implementation of Algorithm 1. For brevity, we use $\mathrm{OPT}_{\mathrm{IUB}}$ as an example. One could increase the optimization problem size of $n \times m$ to $(n \cdot k_b) \times (m \cdot k_r)$ using Algorithm 2, where superscripts indicate matrix shape, left subscripts denote accumulation indices (distinguishing them from element indices).

---

**Algorithm 2** IBPO with Sample Accumulation

---

**Require:** prompt set $\mathbb{D}$, batch size $n$, number of responses $m$, init policy $\pi_0 = \pi_{\mathrm{ref}}$, num of iters $T$, budgets $q_+$

1: **for** $t = 1, \ldots, T$ **do**
2:     **for** $i = 1, \ldots, k_b$ **do**
3:         **prompt sampling**: $_i\mathbf{X}^n \sim \mu_{\mathbb{D}}$
4:         **for** $j = 1, \ldots, k_r$ **do**
5:             **response generation**: $_{ij}\mathbf{Y}^{n \times m} \sim \pi_{\theta_t}(_i\mathbf{X})$
6:         **end for**
7:     **end for**
8:     **prompt accumulation**: $\tilde{\mathbf{X}}^{(n \cdot k_b)} = [_1\mathbf{X}, _2\mathbf{X}, \cdots, _{k_b}\mathbf{X}]$
9:     **response accumulation**: $\tilde{\mathbf{Y}}^{(n \cdot k_b) \times (m \cdot k_r)} = [_{11}\mathbf{Y}, _{12}\mathbf{Y}, \cdots, _{1k_r}\mathbf{Y}; \cdots; _{k_b1}\mathbf{Y}, _{k_b2}\mathbf{Y}, \cdots, _{k_bk_r}\mathbf{Y}]$
10:     **evaluate correctness** and **empirical KL**: $\mathbf{R}_{\mathrm{match}} = r_{\mathrm{match}}(\tilde{\mathbf{X}}, \tilde{\mathbf{Y}})$ and $\hat{\mathbf{KL}} = \mathbb{KL}(\tilde{\mathbf{Y}}; \tilde{\mathbf{X}}, \pi_{\mathrm{ref}})$
11:     **margin maximization**: $\hat{\pi}^\star_{\tilde{\mathbf{X}}, \tilde{\mathbf{Y}}} \in \mathrm{OPT}_{\mathrm{IUB}}(\tilde{\mathbf{X}}, \tilde{\mathbf{Y}}, \mathbf{R}_{\mathrm{match}}, \hat{\mathbf{KL}}, q_+)$ as defined in Eq. (8)
12:     **gradient update**: with $-\sum_{i=1}^{n \cdot k_b} \sum_{j=1}^{m \cdot k_r} \hat{\pi}^\star_{\tilde{\mathbf{X}}, \tilde{\mathbf{Y}}}(y_{ij}|x_i) \frac{\partial}{\partial \theta} \log \pi_\theta(y_{ij}|x_i)\big|_{\theta=\theta_t}$
13: **end for**

---

# D. Sample Responses

We show a couple of example responses of ASV-IuB-$50\%$, the omitted prompt could be found in Figure 4.

---

**Voting Example**

Prompt omitted ...

Problem: Compute $\arcsin\left(-\frac{1}{2}\right)$. Express your answer in radians.

[TRIAL] ## Step 1: Recall the definition of arcsine The function $\arcsin x$ is the inverse of the sine function. It returns the angle whose sine is $x$.
## Step 2: Use the unit circle to find the angle From the unit circle, we know that $\sin\left(-\frac{\pi}{3}\right) = -\frac{1}{2}$.
## Step 3: Determine the value of $\arcsin\left(-\frac{1}{2}\right)$ Since $\sin\left(-\frac{\pi}{3}\right) = -\frac{1}{2}$, we have $\arcsin\left(-\frac{1}{2}\right) = -\frac{\pi}{3}$.

The final answer is: $\boxed{-\dfrac{\pi}{3}}$. [/TRIAL]

[TRIAL] ## Step 1: Recall the sine function values for common angles. The sine of $\frac{\pi}{3}$ is $\frac{\sqrt{3}}{2}$.
## Step 2: Use the relationship between sine and arcsine. Given $\sin(\theta) = -\frac{1}{2}$, we are looking for $\arcsin\left(-\frac{1}{2}\right)$.
## Step 3: Identify the angle whose sine is $-\frac{1}{2}$. From the unit circle or trigonometric identities, we know that $\sin\left(-\frac{\pi}{6}\right) = -\frac{1}{2}$.
## Step 4: Express the answer in radians. Therefore, $\arcsin\left(-\frac{1}{2}\right) = -\frac{\pi}{6}$.

Therefore, the final answer is: $\boxed{-\dfrac{\pi}{6}}$ [/TRIAL]

[TRIAL] ## Step 1: Recall the sine function's relationship with the unit circle. The sine of an angle in a right triangle is the ratio of the length of the side opposite the angle to the length of the hypotenuse. For angles in the unit circle (a circle with radius 1), the sine of an angle is the y-coordinate of the point where the terminal side of the angle intersects the circle.
## Step 2: Identify the angle with a sine of $-\frac{1}{2}$. From the unit circle, we know that $\sin\left(\frac{\pi}{3}\right) = \frac{1}{2}$. Since sine is negative in the third and fourth quadrants, and we are looking for an angle whose sine is $-\frac{1}{2}$, the angle must be in either the third or fourth quadrant.
## Step 3: Determine the angle in the correct quadrant. Considering the unit circle, the reference angle for $\frac{\pi}{3}$ is $\frac{\pi}{3}$ itself. Since we are looking for an angle with a sine of $-\frac{1}{2}$, and knowing that sine is negative in the third and fourth quadrants, the angle we are looking for is in the fourth quadrant because $\sin\left(\frac{5\pi}{3}\right)$ would indeed be negative.
## Step 4: Calculate the angle. The angle whose sine is $-\frac{1}{2}$ is $\frac{5\pi}{3}$.

Therefore, the final answer is: $\boxed{\dfrac{5\pi}{3}}$ [/TRIAL]

[TRIAL] ## Step 1: Recall the definition of arcsine The function $\arcsin x$ is the inverse of the sine function. It returns the angle whose sine is $x$.
## Step 2: Use the sine function to find the angle We want to find the angle whose sine is $-\frac{1}{2}$. We can use the sine function to find this angle.
## Step 3: Recall common angles and their sines From the unit circle or trigonometric identities, we know that $\sin\left(\frac{\pi}{6}\right) = \frac{1}{2}$. Since sine is an odd function, $\sin\left(-\frac{\pi}{6}\right) = -\frac{1}{2}$.
## Step 4: Determine the angle Therefore, the angle whose sine is $-\frac{1}{2}$ is $-\frac{\pi}{6}$.

Therefore, the final answer is: $\boxed{-\dfrac{\pi}{6}}$ [/TRIAL]

[TRIAL] ## Step 1: Recall the sine function's values for common angles. The sine function has well-known values for certain angles. Specifically, $\sin\left(\frac{\pi}{6}\right) = \frac{1}{2}$.
## Step 2: Use the symmetry and periodicity of the sine function. Given that $\sin\left(\frac{\pi}{6}\right) = \frac{1}{2}$, and knowing that sine is an odd function, we have $\sin\left(-\frac{\pi}{6}\right) = -\frac{1}{2}$. This step utilizes the property of odd functions where $f(-x) = -f(x)$.
## Step 3: Solve for $\arcsin\left(-\frac{1}{2}\right)$. Given that $\sin\left(-\frac{\pi}{6}\right) = -\frac{1}{2}$, the angle whose sine is $-\frac{1}{2}$ is $-\frac{\pi}{6}$. This is because the inverse sine function, $\arcsin$, returns the angle whose sine is a given value.

Therefore, the final answer is: $\boxed{-\dfrac{\pi}{6}}$ [/TRIAL]

The answer $\boxed{-\dfrac{\pi}{6}}$ has occured three times and is considered as a consensus. The final answer is $\boxed{-\dfrac{\pi}{6}}$. I hope it is correct.

---

---

**Non-voting Example**

Prompt omitted ...

Problem: If $f(x) = \frac{3x-2}{x-2}$, what is the value of $f(-2) + f(-1) + f(0)$? Express your answer as a common fraction.

[TRIAL] ## Step 1: Substitute $x = -2$ into the function $f(x)$. To find $f(-2)$, substitute $x = -2$ into the function:

$$f(-2) = \frac{3(-2) - 2}{-2 - 2} = \frac{-6 - 2}{-4} = \frac{-8}{-4} = 2.$$

## Step 2: Substitute $x = -1$ into the function $f(x)$. To find $f(-1)$, substitute $x = -1$ into the function:

$$f(-1) = \frac{3(-1) - 2}{-1 - 2} = \frac{-3 - 2}{-3} = \frac{-5}{-3} = \frac{5}{3}.$$

## Step 3: Substitute $x = 0$ into the function $f(x)$. To find $f(0)$, substitute $x = 0$ into the function:

$$f(0) = \frac{3(0) - 2}{0 - 2} = \frac{0 - 2}{-2} = \frac{-2}{-2} = 1.$$

## Step 4: Sum the values of $f(-2)$, $f(-1)$, and $f(0)$. Sum the calculated values:

$$f(-2) + f(-1) + f(0) = 2 + \frac{5}{3} + 1.$$

## Step 5: Simplify the sum to express it as a common fraction. To combine the terms into a common fraction, convert the whole numbers to fractions with a common denominator:

$$2 + \frac{5}{3} + 1 = \frac{6}{3} + \frac{5}{3} + \frac{3}{3} = \frac{6 + 5 + 3}{3} = \frac{14}{3}.$$

Therefore, the value of $f(-2) + f(-1) + f(0)$ is $\frac{14}{3}$.

Therefore, the final answer is: $\boxed{\dfrac{14}{3}}$ [/TRIAL]

Terminated due to difficulty level.

---

# E. Training Curves & Further Discussions

**Budgets constraints.** In Section 5, we show that the constraints hold exactly on the training set, for responses that are correct. Figure 6b further shows the ratio of voting responses for all online samples. It's clear that the constraints also hold.

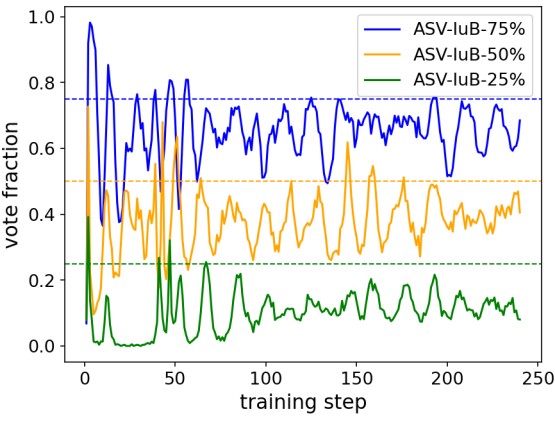 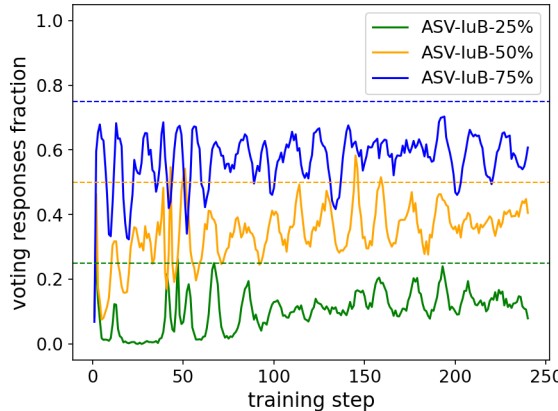

(a) Voting ratio of correct responses (training set)    (b) Voting ratio of all responses (training set)

Figure 6: Voting response ratio versus training steps. Dashed line denotes the budget $q_+$. On the training set, IuB formulation follows the budget constraints almost exactly for both: (a) correct responses; (b) all responses.

**CGPO on $\mathbb{D}_{\text{RL}}$ w/ LLaMA.** To further understand the results we presented in Section 5 is benfit from our budget-aware formulation or from the prompt set of Lai et al. (2024). We further run CGPO with open-sourced LLaMA model and the SDPO dataset in the SCoT format. Figure 7 compares the training dynamics of CGPO with LLaMA versus our ASU-IuB-$q_+$ experiments. In general, the SDPO prompt set does not provide much additional knowledge as suggested by the OSS w/ vanilla CGPO experiment, but ASV-IuB-$q_+$ experiments are able to achieve noticable gain.

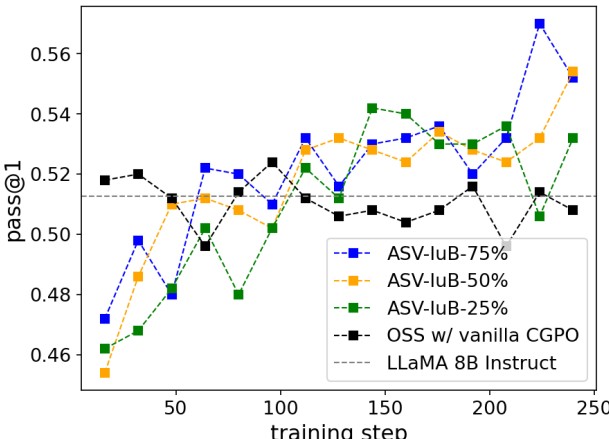

Figure 7: Training curves: each point corresponds an evaluation of MATH500 test set, and dashed line is the pass@1 of LLaMA 8B Instruct.

**Controlled setting.** Our experiment setting is minimal: we do not use reward models; we use only an 8B model to generate any set of sampled responses, $\mathbb{A}^{\text{SAMPLE}}$. Our RL training set, $\mathbb{D}_{\text{RL}}$, contains only 10k prompts, leaving the setting quite controlled and providing room for improvement through engineering efforts.

**Limitations.** Our work is limited in its choice of RL algorithms and applications. While it should be straightforward to apply our framework to different RL frameworks, additional effort is required to derive an optimization problem for different application tasks. Future work may include extending our constrained policy optimization framework to a broader range of LLM applications or scaling up our experiments.

# F. Hyperparameters

We list the hyperparameters used for experiments setup 1.2, 2.1, 2.2, as described in Table 7. And we conduct our experiments with NVIDIA-A100-80Gs. (Please refer to Xu et al. (2024b) for the definition of some RL-specific hyper-parameters.)

Table 8: Hyperparameters for Experiment setups 1.2, 2.1, and 2.2

| Hyperparameter | Setup 1.2 | Setup 2.1 | Setup 2.2 |
|---|---|---|---|
| prompt size | 11295 | 11295 | 10795 |
| number of nodes | 4 | 4 | 8 |
| learning rate | 1e-6 | 1e-6 | 5e-7 |
| batch size (per node) | 8 | 16 | 4 |
| num of steps | 1024 | 2048 | 240 |
| optimizer | AdamW | AdamW | AdamW |
| scheduler | constant | constant | constant |
| packing | yes | yes | - |
| max sequence length | 32768 | 32768 | 6144 |
| gradient accumulation | 1 | 2 | 1 |
| RL-specific params | | | |
| num generation per prompt | | | 8 |
| max generation length | | | 4096 |
| temperature | | | 1.0 |
| top-p | | | 0.9 |
| KL-threashold | | | 1024 |
| batch accumulation $k_b$ | | | 4 |
| response accumulation $k_r$ | | | 1 |

# G. Data Construction

We elaborate on the details of $\mathbb{Q}_{\text{SDPO}}$, $\mathbb{A}_{\text{SDPO}}^{\text{GOLDEN}}$, $\mathbb{Q}_{\text{MATH}}$, $\mathbb{A}_{\text{MATH}}^{\text{SAMPLE}}$, and the filteration function $\mathcal{F}$.

**SDPO dataset.** Lai et al. (2024) create a dataset (see their Section 3.2) contains 10,795 step-wise preference pairs, where the winning response is a correct solution, the losing response is incorrect starting from certain step. For our purpose of SFT and RL training, we do not need losing responses, hence we create $\mathbb{A}_{\text{SDPO}}^{\text{GOLDEN}}$ with only wining (correct) responses. The step-formatted response (see an example below) is the reason we choose the dataset from Lai et al. (2024), as it adhere the SCoT format of LLaMA models. Therefore, one do not need to worry about potential format changes.

> **SDPO question-response example**
>
> Problem: Sides of a rectangular park are in the ratio 3: 2 and its area is 3750 sq m, the cost of fencing it at 40 ps per meter is?
> ## Step 1: Let's assume the sides of the rectangle are $3x$ and $2x$. The area of the rectangle is the product of its sides, so we have $3x \cdot 2x = 6x^2 = 3750$. Solving for $x$, we get $x^2 = \frac{3750}{6} = 625$, so $x = \sqrt{625} = 25$.
> ## Step 2: So the sides of the rectangle are $3x = 3 \cdot 25 = 75$ meters and $2x = 2 \cdot 25 = 50$ meters.
> ## Step 3: The perimeter of the rectangle is $2 \times (75 + 50) = 2 \times 125 = 250$ meters.
> ## Step 4: The cost of fencing is 40 paise per meter, so the total cost of fencing is $250 \times 40 = 10000$ paise. Since 1 rupee is equal to 100 paise, the cost in rupees is $\frac{10000}{100} = 100$ rupees.
> Therefore, the final answer is: $\boxed{100}$

**SV dataset for MATH.** We also created datasets, $\mathbb{D}_{\text{SV}}$ and $\mathbb{D}_{\text{ASV1}}$, for SFT so that a model could follow the SV instructions. We take $\mathbb{D}_{\text{SV}}$ as an example and $\mathbb{D}_{\text{ASV1}}$ could be created similarly. To do so, we first generate 32 responses per prompt for the entire MATH training split with a temperature of 1.2 and top-p of 0.9. We then apply our SV templates $\mathcal{T}_q$ and $\mathcal{T}_a$ to create corresponding SV question and answer pairs. The procedure of creating a SV response is given by Algorithm 3. While we have include an example of SV responses in Figure 8, we make it more concrete the SV response template below,

We then apply some filtration $\mathcal{F}$ to the created dataset. We remove SV responses whose final answers are incorrect. From the remaining set, we sub-sample 500 question-response pairs. These pairs are selected from problems that have between 4 and 8 distinct answers out of 32 samples, ensuring that we construct sequential responses with a diverse trials. The distribution of trial counts and distinct answer counts are shown in Figure 9. It can be observed that the filtered data is diversely distributed.

| **SV template:** $\mathcal{T}_a(A_1, \cdots, A_k, \textbf{final answer})$ **(if consensus found)** | **SV template:** $\mathcal{T}_a(A_1, \cdots, A_8, \textbf{final answer})$ **(if no consensus found)** |
|---|---|
| [TRIAL] $\{A_1\}$ [/TRIAL] 
 [TRIAL] $\{A_2\}$ [/TRIAL] 
 $\cdots$ 
 [TRIAL] $\{A_k\}$ [/TRIAL] 
 The answer ⬚final answer has occurred three times, and is considered as a consensus. 
 The final answer is ⬚final answer. I hope it is correct. | [TRIAL] $\{A_1\}$ [/TRIAL] 
 [TRIAL] $\{A_2\}$ [/TRIAL] 
 $\cdots$ 
 [TRIAL] $\{A_8\}$ [/TRIAL] 
 Maximum trials reached but no consensus found due to a tie; the most plausible answer is ⬚final answer. 
 The final answer is ⬚final answer. I hope it is correct. |

Figure 8: SV response templates. Left: suppose a consensus is found at $k$-th answer; Right: no consensus found. Note the subscript $i$ of $A_i$ only denotes index of answer, $A_i$ and $A_j$ could still have same final answer for $i \neq j$.

---

**Algorithm 3** Creating SV Response for SFT .

---

**Require:** template $\mathcal{T}_q$, $\mathcal{T}_a$, a problem $Q$, a set of shuffled responses $\{A_i : i = 1, 2, \cdots, K\}$
 1: create prompt: $\mathcal{T}_q(Q)$                                           replace problem placeholder with $Q$
 2: responses = []; final_answer = [INVALID_ANSWER]; found = False
 3: **for** $i = 1, 2, \cdots, K$ **do**
 4:     responses.append($A_i$)
 5:     majority = find_majority(responses)
 6:     majority_count = count_majority(responses, majority)
 7:     **if** majority_count == 3 **then**
 8:         found = True; final_answer = majority; break
 9:     **end if**
10: **end for**
11: **if** found == False **and** responses contain correct solution **then**
12:     final_answer = random pick a correct solution
13: **end if**
14: create SV response: $\mathcal{T}_a$(responses, final_answer)

---

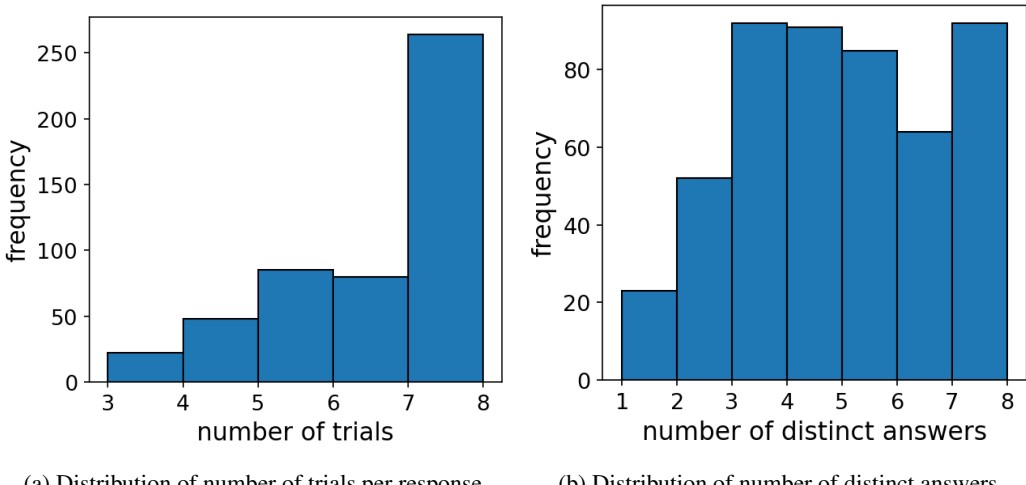

(a) Distribution of number of trials per response        (b) Distribution of number of distinct answers

Figure 9: The distribution of filtered prompt-responses subset, which suggests that the construct data is generally diversely distributed.

# H. Integer Linear Program Solving

We use SciPy MILP solver, available here, to solve an integer LP every iter. Figure 1 is a Pythonic pseudo-code. And the wall-time consumed could be found in Figure 10. As the problem size is small, so the computational overhead is negligible.

```python
import numpy as np
from scipy.optimize import milp, LinearConstraint, Bounds

def solve_iub(acceptance, is_vote, budget):
    """ solves a Inference under Budget (IuB) problem. Parameters:
    - acceptance: an n x m array where each element is 1 if accepted, 0 otherwise.
    - is_vote: an n x m array indicating whether response is voting (1) or non-voting (0).
    - budget: the fractional budget (q+) constraint for the problem. """
    n, m = acceptance.shape
    # calculate pass rates for vote-based and non-vote-based responses
    vote_pass_rate = np.mean(acceptance * is_vote, axis=1, keepdims=True)
    sample_pass_rate = np.mean(acceptance * (1 - is_vote), axis=1, keepdims=True)
    margin = vote_pass_rate - sample_pass_rate   # (n, 1)
    # flattern acceptance and vote indicator, tile the margin
    acceptance = np.reshape(acceptance, -1)               # (n x m, )
    is_vote = np.reshape(is_vote, -1)                     # (n x m, )
    margin = np.reshape(np.tile(margin, [1, m]), -1)     # (n x m, )

    # define the objective function coefficients
    c = -1 * margin * is_vote + margin * (1 - is_vote)
    # acceptance constraints: ensure each prompt meets acceptance criteria
    A_acceptance = np.eye(len(acceptance))
    b_acceptance = acceptance
    # one response per problem constraint (BoN)
    A_problem = np.zeros((n, n * m))
    for i in range(n):
        A_problem[i, i * m:(i + 1) * m] = 1
    b_problem = np.ones(n)
    # voting responses budget constraint
    A_vote_budget = np.where(is_vote == 1, 1, 0).reshape(1, -1)
    vote_budget = np.round(budget * len(acceptance))

    # combine all constraints into a single matrix
    A = np.vstack([A_acceptance, A_problem, A_vote_budget])
    b_lower = -np.inf * np.ones(A.shape[0])   # lower bounds for constraints
    b_upper = np.hstack([b_acceptance, b_problem, vote_budget])   # upper bounds
    # solve the MILP problem using the defined objective and constraints
    result = milp(c, integrality=np.ones(len(c)), bounds=Bounds(0, 1),
                  constraints=LinearConstraint(A, b_lower, b_upper))
    return result
```

Listing 1: Pythonic code snippet for solving IuB with SciPy: Note that this is for demonstration purposes, and error-free execution is not guaranteed, due to omitted corner cases.

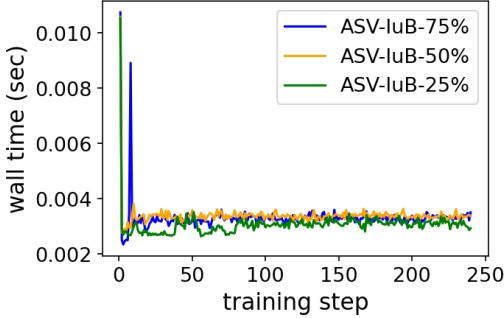

Figure 10: Wall time (averaged across ranks) spent by solver in seconds.

