# OpenReview forum: "Think Smarter not Harder: Adaptive Reasoning with Inference Aware Optimization"
_ICML.cc/2025/Conference — ICML 2025 poster_

### Official Review · Reviewer_8P46 · 2025-03-12

**Overall Recommendation:** 4

**Summary:**

This paper targets the problem of solving mathematical problems with LLMs. While the currently prevalent method LongCoT brings promising improvements for mathematical reasoning, they are sometimes unnecessary and cause token waste. To alleviate this issue, the authors propose an algorithm Inference Budget-Constrained Policy Optimization (IBPO), where the problem is formulated as a resource allocation scenario. The budget is assigned with respect to the difficulty level of the problem. IBPO is implemented based on the RL objective of constraint generative policy optimization by replacing it with margin maximization under budget. Experiments on MATH dataset verify the effectiveness of the proposed method. Analyses also prove its design motivation, that difficult problems will get more budget to solve.

**Claims And Evidence:**

The major claims in the paper, specifically regarding the motivation and experiment results, are good to me.

**Essential References Not Discussed:**

N/A

**Experimental Designs Or Analyses:**

The overall experiment design makes sense. I have some minor issues or comments:
- The authors demonstrate (specifically in Figure 2 Column 3) that the voting budget can adaptively change w.r.t. the difficulty level of the problem. Apart from the token budget, I am wondering if the authors could also offer accuracy on different difficulty levels. Now there is only overall accuracy in Table 3.
- The design choices of the dataset construction in Section 4 (Appendix B) need more justification. For example, why at most "8 trials" and early stop if an answer appears "3 times"? Would the design choices have a significant impact on the final performance (sensitivity)?

**Methods And Evaluation Criteria:**

The paper is evaluated mainly on MATH dataset, which contains annotations on the difficulty levels of different problems. This setting is natural and could well validate the proposed method, as it claims to allocate different budgets on different questions.

**Other Comments Or Suggestions:**

See the above review.

**Other Strengths And Weaknesses:**

See the above review.

**Questions For Authors:**

See the above review.

**Relation To Broader Scientific Literature:**

This paper studies a very important and interesting research question of the current community, allowing LLMs to allocate different token budgets for different problems of difficulty levels. This is timely research, especially when o1 and Deepseek styles begin to dominate in reasoning tasks.

In terms of ideas and findings, many concurrent works try to decrease the number of tokens as much as possible while maintaining performance, such as TokenSkip, this paper takes the approach of resource allocation and designs a novel algorithm to do this. While a lot of previous works noticed the correlation between token budget and problem difficulty, they are mostly doing direct preference alignment. The algorithm proposed in this paper is valuable.

**Theoretical Claims:**

I checked the proposed algorithm and the corresponding equations, all seem sound to me.

---

> ### Author Rebuttal · Authors · 2025-03-31
>
> Thank you very much for your thorough review and valuable feedback! We greatly appreciate your insightful suggestions and positive assessment!
>
> ---
>
> > The authors demonstrate (specifically in Figure 2 Column 3) that the voting budget can adaptively change w.r.t. the difficulty level of the problem. Apart from the token budget, I am wondering if the authors could also offer accuracy on different difficulty levels. Now there is only overall accuracy in Table 3.
>
> | Level               | 1             | 2              | 3              | 4              | 5              | Overall       |
> |--------------------|---------------|----------------|----------------|----------------|----------------|----------------|
> | # of problems | 43            | 90             | 105            | 128            | 134            | 500            |
> | q₊ = 0.25          | 93.02% (40)   | 76.67% (69)    | 64.76% (68)    | 48.44% (62)    | 23.88% (32)    | 54.2% (271)    |
> | q₊ = 0.5           | 83.72% (36)   | 80.00% (72)       | 66.67% (70)    | 52.34% (67)    | 23.88% (32)    | 55.4% (277)    |
> | q₊ = 0.75          | 90.70% (39)    | 77.78% (70)    | 74.29% (78)    | 51.56% (66)    | 23.88% (32)    | 57.0% (285)      |
>
> Absolutely. We break down the overall accuracies reported in Table 3 by difficulty level in the table above (numbers in parentheses indicate the number of problems solved). Some observations are:
>
> - With larger budgets, we observe improved performance on the harder levels (3 and 4), as more voting budget becomes available.
>
> - It is surprising that level 5 performance remains the same across all budgets. We conjecture that this is due to the limitations of the 8B model, which may only solve a subset of the hardest problems.
>
> ---
>
> > The design choices of the dataset construction in Section 4 (Appendix B) need more justification. For example, why at most "8 trials" and early stop if an answer appears "3 times"? Would the design choices have a significant impact on the final performance (sensitivity)?
>
> In general, these choices do impact the performance of sequential voting (SV) itself. These choices were made to ensure that SV performs **comparably** to majority voting (MV), a baseline that is more familiar to the community.
>
> - **Reason of chocies**:
>   - In our observations, SV without early stopping tends to underperform compared to parallel MV. With the current setup, SV achieves performance similar to MV (Figures 2a/2b), which allows for a meaningful comparison between our ASV-IuB-$q$ models and MV.
>
>   - We set the trial cap at 8 to avoid unbounded responses. This limit, combined with early stopping, empirically matched MV performance, so we kept it.
>
> Since our key insights lie in the RL component, we found that aligning SV with parallel MV is a reasonable choice, as it allows for: (i) a sufficient reward margin for optimization, and (ii) a fair comparison to a community-standard baseline (MV).
>
>
> ---
>
> Once again, we gratefully thank the reviewer for the insightful suggestions and positive evaluation! We hope our clarifications have adequately addressed your questions. We sincerely appreciate your support and will carefully revise the manuscript to further improve its clarity!

---

### Official Review · Reviewer_hh45 · 2025-03-13

**Overall Recommendation:** 2

**Summary:**

This paper proposes IBPO to optimize reasoning length allocation in large language models (LLMs). While extended reasoning chains improve accuracy, they often lead to inefficiencies by applying unnecessary long reasoning to trivial queries. IBPO formulates this as a constrained reinforcement learning (RL) problem, categorizing responses into different length groups and imposing density constraints to allocate inference budgets adaptively.

Empirical results show that IBPO improves efficiency, achieving accuracy gains on MATH500 over LLaMA3.1 8B Instruct, with efficiency gains compares to self-consistency. The paper details IBPO’s derivation, implementation and experiments.

**Claims And Evidence:**

Yes. The claims made in the submission are supported by clear and convincing evidence.

**Essential References Not Discussed:**

To the best of my knowledge, there are no essential related works that are missing from the citations or discussion in the paper.

**Experimental Designs Or Analyses:**

Yes, I checked. I think they have no issues.

**Methods And Evaluation Criteria:**

Yes

**Other Comments Or Suggestions:**

(1) The authors should simplify the formulation appropriately to improve the readability of the paper.
(2) The authors should supplement the paper with experiments comparing the inference time overhead of MV, SV, and ASV to provide a more comprehensive evaluation.

**Other Strengths And Weaknesses:**

Strengths:

(1) The method is rooted in constrained reinforcement learning (RL) and resource allocation theory, providing a rigorous framework for optimizing inference budgets.
(2) IBPO aims to balancing inference cost and reasoning accuracy by dynamically adjusting reasoning length based on problem difficulty, reducing unnecessary computational overhead.

Weaknesses:

(1) The paper frequently refers to extended CoT (or long CoT) but selects SV as the representative for extended CoT. This deviates significantly from the standard academic understanding, where extended CoT is typically represented by models like O1 and R1, which focus on deeper rather than broader reasoning (e.g., majority voting). Since the proposed method has not been evaluated on R1 (O1)-like extended CoT, its effectiveness in such settings remains uncertain.
(2) According to my understanding, SV generate different trails sequentially, considering the quadratic complexity in transformer-based models, SV can result in longer inference times compared to Majority Voting (MV), where trails are generated in parallel. What’s more, the paper uses trails per response as the inference budget metric, which has two key issues:
1. Trail count does not directly correspond to token count, whereas token budget would be a more practical and meaningful measure.
2. Despite a lower token budget, the sequential nature of SV may lead to longer inference times compared to parallel methods like MV. Additional runtime experiments comparing MV, SV, and ASV should be conducted to verify whether the proposed method increases inference time. If the approach actually extends inference time, its practical value is questionable.
(3) While the core idea of IBPO is concise, the formulation in the paper contains substantial redundancy, which hinders readers from extracting the key information efficiently.
(4) In Table 3, the IBPO experiments are conducted on LLaMA3.1-8B, whereas S3C and Self-Refine utilize multiple models different from LLaMA3.1-8B. This inconsistency weakens the rigor of the comparison, making the evaluation less reliable.

**Questions For Authors:**

No other question for authors, please see the weaknesses.

**Relation To Broader Scientific Literature:**

This work contributes to the broader scientific discourse on reasoning in large language models (LLMs), particularly in mathematical problem-solving and efficient inference. It builds on foundational research in chain-of-thought (CoT) prompting, which demonstrates that decomposing reasoning into explicit steps improves problem-solving accuracy. While CoT and its extensions—such as self-correction and multi-turn reasoning.

**Theoretical Claims:**

Yes, I think they have no issues.

---

> ### Author Rebuttal · Authors · 2025-03-31
>
> We sincerely appreciate your time in assessing our paper and your thoughtful feedback.
>
> ---
>
> We’d like to first clarify a possible **misunderstanding** regarding the **role of sequential voting** (SV): it is **not** intended as a significant contribution of this work.
>
> **Role of SV:** It is a **simple, constructive alternative** to o1/R1-style long CoT, with **extended length** and **reasonable performance gains** over conventional CoT.
>
> - **Why an alternative**: As o1 is proprietary and R1 was released on Jan 20, ~10 days before the submission deadline.
>
> - **Why SV**:
>   - **Simple**: A minimal construction that allows us to focus on our RL contribution without the distraction of engineering long responses;
>   - **Characteristics**: (i) gains over standard CoT; (ii) extended length.
>     - SV resembles longCoT in these characteristics (though its gains over CoT are moderate), enabling us to demonstrate the effectiveness of our RL algorithm.
>
> - **What do we expect from SV?**:
>
>   - **Superior accuracy?**: No. As shown in Fig 2, SV performs comparably to parallel majority voting (MV). We expect advanced reasoning methods to outperform MV, and hence SV.
>
>   - **Practical value?**: No. The main insights we convey do **not** lie in SV. After all, it is a dummy construction used to highlight our RL algorithm, which carries the core insights of this work.
>
> We understand the concern that SV may not fully capture how o1/R1 works fundementally. Due to the **unavailability** of R1, we construct SV that at least resembles longCoT in terms of the aforementioned characteristics.
>
>
> Given the considerations above, we believe our construction is a fair and reasonable choice for **illustrative** purposes.
>
>
>
>
> ---
>
> To reiterate, our core contributions are:
> - (i) casting adaptive reasoning as constrained inference;
> - (ii) designing a constrained RL algorithm — which merits particular emphasis — that is simple and grounded.
>
> In addition, the final **accuracy** stems from **two orthogonal axes**: (i) **the reasoning axis**; and (ii) **the RL axis**.
>
> $$
> \underbrace{\text{constraint satisfaction} \to \text{adaptive allocation}}_{\text{the RL axis}} \to \text{final accuracy} \leftarrow \text{SV (the reasoning axis)}
> $$
>
> The derivation of our algorithm and the ''upstream'' metric (constraint satisfaction) are theoretically **agnostic** to the choice of long-response.
>
> We intentionally kept the reasoning axis simple and did **not** attempt to boost performance through it, so as to isolate and highlight the RL component.
>
> These points collectively suggest that a reasonable construction is sufficient for emphasizing our RL contribution.
>
> ---
> We hope these discussions clarify our contributions. We believe our RL contribution—unanimously considered as **sound and novel** by other reviewers—merits recognition.
>
> We quote reviewer toYu:
>
> > not only sheds new lights on adaptive reasoning but also provides a valuable algorithmic contribution to constrained RL as a whole.
>
> ----
> For specific comments:
>
> 1. **R1/O1 type models**: We hope the above clarifies our choice of SV, given the unavailability of such models.
>
> 2. **Trials as metric**:
>    1. - This choice was mainly motivated by Table 3. Since the self-correction works don't report token counts, we use the number of trials/turns as a proxy.
>
>       - Besides, SV-SFT performs similarly to MV in trials (Figures 2a/2b); and scales comparably in tokens (Figure 5b, Appendix). Hence, trials serve as a reasonable proxy.
>
>    2. **Practical value considering inference time**: This is an inherent limitation of any long-form reasoning method like longCoT, not a consequence of our RL algorithm. Substituting SV with O1/R1-style responses would result in similar inference-time overhead.
>
>       After all, SV is a dummy alternative to longCoT, and was **not** intended to offer practical value beyond its illustrative purpose.
>
> 3. **Table 3**: As noted in Sec. 5.1, Table 3 is not intended to suggest our method outperforms self-correction. These are two **orthogonal** research directions: constrained inference and self-correction.
>
>     Its purpose is simply to illustrate that constrained inference can achieve **comparable** performance to **a well-established line of work**. For this purpose, we believe transcribing their results is reasonable—it is convenient and avoids potential discrepancies from re-implementation.
>
>     A more informative comparison is Fig. 2, where we evaluate against an **efficiency boundary** interpolating between two extremes of non-adaptive (homogeneous) cases. See our response to reviewer JCKH for details.
>
> ---
> Again, we sincerely appreciate your suggestions and your assessment that our claims are supported, our theoretical and experimental designs have no issues, and our framework is rigorous.
>
> We will carefully revise the manuscript to further improve clarity based on your suggestions, and we hope our responses have adequately addressed your questions.

---

### Official Review · Reviewer_JCKH · 2025-03-14

**Overall Recommendation:** 2

**Summary:**

This paper discusses a method for scaling LLM test time compute on adaptive basis based on prompt difficulty. The proposed approach is a novel reinforcement learning technique that allocates more inference to difficult problems (adaptive number of votes - where each vote requires an inference) and fewer votes to easy questions. The proposed method, called Inference Budget-Constrained Policy Optimization (IBPO) is experimentally verified to improve performance on MATH500 dataset - though at a higher test time compute budget (2x).

## update after rebuttal

I thank the authors for their response.

While the response partially addresses my concerns, I am still not confident that the approach generalizes beyond the specific datasets and architectures.

As the authors mentioned, the method is applicable for reasoning settings that are neither easy (85% or more accuracy per model) nor hard (AIME where model accuracy is low). Normally, this would not be a problem, one could adjust the capacity of the model (use smaller / larger LLMs) to show that the approach can be performant at these settings, but the authors mentioned that their approach cannot scale well due to training complexity.

On the other hand, while the authors presented an argument as to why the technique should not be compared with evolved SC variants, I am not convinced. If their approach is orthogonal to the reasoning method, then I would like to see some experiments where more modern reasoning backbones are used when combined with their method.

I am certain that more evolved reasoning baselines such as the ones I recommended need to be compared to, to quantify the method's impact on total token usage versus a simple no fine tuning method.

For this reason I am maintaining my current score.

**Claims And Evidence:**

- IBPO can correctly discriminate easy and hard problems (verified in experiments section e.g. Fig. 2)
- IBPO can allocate resources efficiently by dedicating more inference compute towards hard questions (this is theoretically expected from the way the RL objective is set up and empirically verified to do so.
- The paper claims that its proposed method has an improvement that is in relative terms twice that of vanilla Self Consistency (SC) (I am not actually sure where this is verified. The claim is plausible, but I cannot pinpoint where the 2x efficiency is shown - perhaps I am missing it as the paper is quite dense).

**Essential References Not Discussed:**

I understand that [1] was mentioned in the introduction section of the paper but I believe that it merits experimental comparison.

[1] Let’s Sample Step by Step: Adaptive-Consistency for Efficient Reasoning and Coding with LLMs

[2] Escape Sky-high Cost: Early-stopping Self-Consistency for Multi-step Reasoning

**Experimental Designs Or Analyses:**

1. There does not seem to be a sufficient amount of experiments. Evaluating on MATH 500 is decent choice, but why not on other datasets? Other datasets with mathematical reasoning problems include AIME 2024 (or other years of AIME), SVAMP, ASDIV, AQUA, some big bench hard datasets, etc.
2. I am curious as to why the selected LLMs seems to be small. The experiments go up to an 8B model. Do the performance gains generalize across different parameter sizes? What about other LLM families besides LLAMA?
3. Baseline selection: In the abstract the authors make a point of improving over vanilla self consistency (SC). I am curious if the proposed method competes well against modern SC variants that have been demonstrated to significantly reduce inference costs. Given that vanilla SC is reasonably dated I think it would be fair to compare against [1] and [2].

[1] Let’s Sample Step by Step: Adaptive-Consistency for Efficient Reasoning and Coding with LLMs

[2] Escape Sky-high Cost: Early-stopping Self-Consistency for Multi-step Reasoning

**Methods And Evaluation Criteria:**

1. The RL method seems correct and novel. I would like to note that while I have understanding of RL, I am not an RL expert thus I would defer to other reviewers to validate my positive evaluation of the proposed method.
2. Evaluation metrics are reasonable. The experimental pipeline is a fair way test for the method.

**Other Comments Or Suggestions:**

N/A

**Other Strengths And Weaknesses:**

This is polished paper but I personally find it dense to read with a lot of heavy notations. Perhaps it could be made sparser with more references to appendix to make it easier? For example, it was not easy to understand the dataset preparation section.

**Questions For Authors:**

- In table 3 it is stated that a lot of the numbers are duplicated from Qu et al. (2024); Yan et al. (2024); Kumar et al. (2024). Did the authors run the code for some of these methods in their setting to make sure that the results reproduce? Is the setting reported from these works identical? For example, are the exact same prompts used in all works in this table?

**Relation To Broader Scientific Literature:**

The work relates to the test time compute literature. This is a major area of research for LLMs. The paper's problem is well motivate.

**Theoretical Claims:**

N/A

---

> ### Author Rebuttal · Authors · 2025-04-01
>
> Thank you for your insightful feedback!
>
> ---
> We extend some discussion on **accuracy** to clarify our evaluation design, which also helps explain our choice of baseline later on.
> While important, accuracy depends on **two orthogonal axes**: (i) the RL axis; and (ii) the reasoning axis.
> $$
> \underbrace{\text{constraint satisfaction} \to \text{adaptive allocation}}_{\text{RL}} \to \text{accuracy} \leftarrow \text{SV (reasoning)}
> $$
> - **RL axis**: We're glad the reviewer agrees the adaptive allocation is ''**theoretically expected**'' and ''**verified**'';
> - **Reasoning axis**: (A)SV was intended as a **simple** alternative to longCoT with **reasonable** accuracy, as R1 was unavailable until 10 days before the submission deadline. Empirically, (A)SV performs **comparably** to SC (Fig 2), enabling:
>      -  a sufficient margin for optimization;
>      -  a fair comparison w/ SC, a baseline more familiar to the community.
>
> (SC is used here to match the reviewer’s terminology, though we use MV in paper.)
>
> One key observation is: (i) SV is an all-voting case (can be seen as $q_+ = 1$); (ii) ours are adaptive with voting capped by $q_+$. This suggests, roughly speaking, that ours should be theoretically **upper bounded** by SV.
>
> Since SV performs comparably to SC, we consider SC a reasonable baseline—based on the reasons above, not chosen ad hoc.
>
> While further engineering on the reasoning axis (e.g. better alternative than SV) could improve overall accuracy, we **refrained** from over-optimizing it, as it's **not** the core insight of this work and could distract from the RL contributions.
>
> ---
> - **Baseline**: Continuing from the discussion above
>
>   - **Interpolation baseline**: We compare ours against a **hypothetical efficiency boundary (HEB)** (gray line Fig 2), defined by two **non-adaptive** extremes:
>     - **$q_+ = 0$**: All short responses;
>     - **$q_+ = 1$**: All long responses.
>
>     A model **above** the HEB can be interpreted as having **improved efficiency**.
>
>   - **Why SC**: SC was chosen because it closely aligns with the HEB (Fig 2). We mention SC in the abstract, as readers are likely more familiar with it.
>
>   - **[1, 2]**: While [1, 2] are more recent, our goal is to evaluate against the HEB, which reflects non-adaptive cases. As we refrained from over-optimizing the reasoning axis, the upper bound (SV ≈ SC) is relatively low—but sufficient for testing our RL part—and is likely outperformed by [1, 2]. Thus, we don’t find a direct comparison with them entirely fair/necessary.
>
>   We hope it clarifies our baseline choice and will revise accordingly, including additional discussion of [1, 2].
>
> - **Dataset**: MATH500: (i) it has difficulty metadata; (ii) LLaMA 3.1 8B Inst. has moderate accuracy. All numbers hereafter refer to this model.
>
>     Apart from metadata: Easier ones (GSM8K, SVAMP, ASDIV w/ pass@1 of ~85% [3,4,5]) have small reward margins, as most queries are solved in one attempt;
>     For harder ones like AIME (2/30 solved [6]), efficiency makes little sense as the model rarely succeeds.
>
>     We hope our grounded derivation and constraint satisfaction curves offer more insight and confidence than accuracy numbers alone.
>
> - **Model**: We don't particularly find 8B ''small'', especially given the cost of online training. For reference, [7]—an online self-correction—uses 7B models.
>   While we didn't explore other families, we're optimistic about the method's generalizability, thanks to its (i) model-agnostic derivation; (2) simple update (Eq. 5).
>
> ---
> - **2x efficiency**: Apologies for the confusion. It comes from Tab 3 and Fig 2. At $q_+ = 50%$, ours has 55.4% (a 4.14% gain) w/ 2.16x trials (Tab 3), while **the HEB** (aligned with SC, if not better) in Fig 2 requires ~4.5x trials for similar accuracy—implying ~2x efficiency.
>
> - **Table 3**: We didn't re-run; the numbers were taken from the cited papers to avoid implementation discrepancies.
>
> ---
> - Online training is **inherently** expensive: For 70B, under our setup, we estimate that a single epoch of generation alone could take ~2,600 H100 hours at 300 output tokens/sec per node. In practice, training throughput is significantly lower due to memory overhead from model optimization—and this estimate excludes other potential costs. These limited us from exploring larger models and harder datasets (which likely require larger training sets).
>
> ---
> We quote reviewer toYu:
> > not only sheds new lights on adaptive reasoning but also provides a valuable algorithmic contribution to constrained RL as a whole.
>
> We are not aware of RL-for-LLM works that explicitly impose linear constraints to control response distributions.
>
> We believe our RL contribution merits stronger recognition—especially given your positive evaluation of it.
>
> ---
> Again, we appreciate your valuable suggestions! We will revise accordingly and we hope our responses addressed your questions.
>
> ---
> [3] arxiv.org/abs/2407.21783
>
> [4] 2410.06638
>
> [5] 2502.12134
>
> [6] 2410.01560
>
> [7] 2407.18219

---

### Official Review · Reviewer_toYu · 2025-03-23

**Overall Recommendation:** 4

**Summary:**

With the prevalence of Chain of Thought (CoT) in complex reasoning and the emergence of ultra-long reasoning models such as OpenAI-o1 and DeepSeek-R1, unnecessarily tedious and long generations for trivial problems are increasingly becoming a problem. The paper approaches this problem from a RL perspective, proposing IBPO (Inference Budget-Constrained Policy Optimization) that, rather than simply taking a metric of context length as reward, forms a constraint-RL by controlling how response lengths are distributed. The algorithm is then deduced to be a generalization of iterative SFT. Experiments and positive results are shown.

**Claims And Evidence:**

A few claims are made during the deduction of main algorithm, but they're all quite well-grounded.

When finding  a workaround for solving the parametric objective function with constraints, authors claim that the non-parametric workaround is superior to alternating between gradients of the policy and Lagrangian multipliers. Although no ablation studies are provided, solid literature on other methods utilizing this workaround in similar problems supports the claim well.

It's also claimed early on in the paper that the proposed IBPO ends up becoming a generalization of SFT methods which is confusing at first but natural after the table comparing SFT, RFT, RAFT and IBPO is provided.

The choice of some techniques, however, weren't explicitly analyzed/compared, such as the use of semi-gradient and the choice of CGPO, but they are either intuitive choices or self-explanatory in positive experiment results.

**Essential References Not Discussed:**

None that I'm aware of.

**Experimental Designs Or Analyses:**

Extensive experiments were conducted, comparing prompt constructing methods and optimizing methods. Experiments consist of the SFT/Prompting-based comparisons and Online Iterative SFT/RL comparisons, respectively revealing effectiveness of the proposed Sequential Voting and IBPO optimization paired with Adaptive Sequential Voting, although improvements seem a little marginal.

As for the setting, extensive baselines were introduced/re-implemented to show the proposed methods' superiority, among which adapting self-correction methods as baseline is a creative choice, as it generates similar long context responses to SV.

**Methods And Evaluation Criteria:**

The main method IBPO is framed initially as a constraint RL problem, but after practical adaptations with respect to the policy update (solving for approximate optimal policy via sampling, using semi-gradient) and the reward calculation (introducing a marginal reward, implementing CGPO), the method is deduced to a SFT where samples take the form of weights assigned to long or short context responses. This is in theory sound and intuitive, except that dichotomy between long and short responses is a little counter-intuitive.

The evaluation is largely sound: authors developed Sequential Voting prompting to generate long-context responses, SCoT for solely short-context responses, and Adaptive Sequential Voting prompting for mixed responses. These are used separately as SFT sets and results are compared. However, I'm still hesitant to accept that responses constructed this way are actually long/short-context responses -- the long ones look like a mere aggregation of short responses.

**Other Comments Or Suggestions:**

On the paper structure, maybe it's a better idea to include more explanation on prompt generation (the proposed SV, ASV), not only to clarify the experiment setting more, but to also convince readers that SV is adequate in representing long response situations that the paper initially tries to improve. The algorithm and experiment parts can be cut down a bit.

**Other Strengths And Weaknesses:**

Aside from previous comments, the paper is thorough and detailed in introducing the algorithm, providing sufficient citation and overview of related literature for most readers to understand.

One potential weakness is that many of the baseline results are not run but transcribed from others' works. This harms the solidity of the experiment comparisons.

**Questions For Authors:**

One major confusion is whether prompts generated via SV sufficiently represent the "unnecessarily tedious long reasoning trace" that the paper sets out to improve. This is rather important as SV seems to be the only experiment scenario.

**Relation To Broader Scientific Literature:**

This paper not only sheds new lights on the problem of budget-aware/adaptive reasoning but also provides a valuable algorithmic contribution to constrained RL as a whole. During parameter update, a method that is essentially a generalization of RFT and RAFT is proposed; the algorithm is a modified version of CGPO, although the alteration is simple.

**Theoretical Claims:**

The theoretical proportion is mainly definitions, intuitive deductions and plugging-ins. They seem correct to me.

---

> ### Author Rebuttal · Authors · 2025-03-31
>
> Thank you very much for your thorough review and valuable feedback! We greatly appreciate your insightful suggestions and positive assessment!
>
> ---
>
> We address these three comments collectively, given their relevance to one another.
>
> > However, I'm still hesitant to accept that responses constructed this way are actually long/short-context responses -- the long ones look like a mere aggregation of short responses.
>
> > One major confusion is whether prompts generated via SV sufficiently represent the "unnecessarily tedious long reasoning trace" that the paper sets out to improve. This is rather important as SV seems to be the only experiment scenario.
>
> > On the paper structure, maybe it's a better idea to include more explanation on prompt generation (the proposed SV, ASV), not only to clarify the experiment setting more, but to also convince readers that SV is adequate in representing long response situations that the paper initially tries to improve.
>
> We understand the concern that SV may not fully capture how o1/R1 works at a fundamental level. This construction was chosen due to the unavailability of R1, which was released on Jan 20—roughly 10 days before the submission deadline.
>
> Reproducing o1/R1-style longCoT, or anything more similar, was beyond the scope of our focus on constrained LLM inference. Moreover, it was unclear to the community how to reproduce such methods until the R1 technical report [1] was released on Jan 22.
>
> The SV construction resembles two key characteristics of longCoT: (i) improved accuracy, and (ii) extended response length. It therefore serves an illustrative purpose to highlight our core RL contribution.
>
> Additionally, our paper (i) casts the problem in terms of distribution constraints, rather than directly optimizing over the long-response group, and (ii) derives an RL method that is theoretically agnostic to the specific type of long response. These efforts together further support the use of a constructed alternative.
>
> Given these considerations—and the fact that o1/R1-style longCoT was not available at the time—we hope you find the use of this construction reasonable.
>
> We completely agree that devoting more space to explaining the design decisions and clarifying the role of SV would significantly improve the paper's clarity. Thank you for pointing this out—we will revise the manuscript accordingly!
>
>
> [1] DeepSeek-R1: Incentivizing Reasoning Capability in LLMs via Reinforcement Learning. https://arxiv.org/abs/2501.12948
>
> ---
>
> > revealing effectiveness of the proposed Sequential Voting and IBPO optimization paired with Adaptive Sequential Voting, although improvements seem a little marginal.
>
> > One potential weakness is that many of the baseline results are not run but transcribed from others' works. This harms the solidity of the experiment comparisons.
>
> We believe these are relatively minor points, if our understanding is correct. As such, we've kept our response concise to avoid unnecessary distraction. (Further details can be found in our response to Point 3 for Reviewer hh45.)
>
> After all, our work on constrained LLM inference and the self-correction literature are essentially two orthogonal directions. Transcribing their results is both convenient and helps avoid potential implementation discrepancies.
>
>
> ---
>
> Once again, we sincerely appreciate the reviewer’s insightful feedback and encouraging assessment! We hope our clarifications sufficiently address your questions and clarify our choice of construction. We deeply appreciate your support and will carefully revise the manuscript to further improve its clarity!

---

> > ### Comment · Reviewer_toYu · 2025-04-05
> >
> > I'd like to express gratitude to the authors for carefully addressing all of my concerns. I understand that SV only serves illustrative purposes, but I believe it's different from longCoT in both the ways it improves accuracy and extends response length, not to mention other significant differences such as inference cost. There's also concerns raised by other reviewers that I find reasonable. Therefore, I maintain my original rating.

---

### Decision · Program_Chairs · 2025-05-01

**Decision:**

Accept (poster)

**Comment:**

This paper introduces IBPO, a reinforcement learning algorithm that helps LLMs assign longer reasoning chains to harder problems and shorter ones to easier ones. The approach frames this as a constrained optimization problem, where the model learns to balance accuracy and computation cost. The method is theoretically sound and the use of constrained RL is interesting. However, the experiments could be more complete by comparing to modern variants of self-consistency methods. Another issue is to use SV to represent models such as o1 and R1 where SV focuses on breath and in fact, o1 and R1 focuses on the depth of the reasoning. The experiments are also limited to the math500 dataset.